# SARS-CoV-2 neutralizing human recombinant antibodies selected from pre-pandemic healthy donors binding at RBD-ACE2 interface

Federico Bertoglio [1,10], Doris Meier[1,10], Nora Langreder[1,10], Stephan Steinke [1,10], Ulfert Rand [2,10], Luca Simonelli[3,10], Philip Alexander Heine[1], Rico Ballmann[1], Kai-Thomas Schneider[1], Kristian Daniel Ralph Roth[1], Maximilian Ruschig[1], Peggy Riese[1,2], Kathrin Eschke[2], Yeonsu Kim[2], Dorina Schäckermann[1], Mattia Pedotti[3], Philipp Kuhn[4], Susanne Zock-Emmenthal[5], Johannes Wöhrle [6], Normann Kilb[6], Tobias Herz[6], Marlies Becker[1], Martina Grasshoff[7], Esther Veronika Wenzel[1], Giulio Russo [1], Andrea Kröger[7], Linda Brunotte [8], Stephan Ludwig [8], Viola Fühner[1], Stefan Daniel Krämer [6], Stefan Dübel [1], Luca Varani[3,11✉], Günter Roth[6,11✉], Luka Čičin-Šain [2,9,11✉], Maren Schubert[1,11✉] & Michael Hust [1,11✉]

COVID-19 is a severe acute respiratory disease caused by SARS-CoV-2, a new recently emerged sarbecovirus. This virus uses the human ACE2 enzyme as receptor for cell entry, recognizing it with the receptor binding domain (RBD) of the S1 subunit of the viral spike protein. We present the use of phage display to select anti-SARS-CoV-2 spike antibodies from the human naïve antibody gene libraries HAL9/10 and subsequent identification of 309 unique fully human antibodies against S1. 17 antibodies are binding to the RBD, showing inhibition of spike binding to cells expressing ACE2 as scFv-Fc and neutralize active SARS-CoV-2 virus infection of VeroE6 cells. The antibody STE73-2E9 is showing neutralization of active SARS-CoV-2 as IgG and is binding to the ACE2-RBD interface. Thus, universal libraries from healthy human donors offer the advantage that antibodies can be generated quickly and independent from the availability of material from recovering patients in a pandemic situation.

[1] Technische Universität Braunschweig, Institut für Biochemie, Biotechnologie und Bioinformatik, Abteilung Biotechnologie, Braunschweig, Germany. [2] Department of Vaccinology and Applied Microbiology, Helmholtz Centre for Infection Research, Braunschweig, Germany. [3] Institute for Research in Biomedicine (IRB), Università della Svizzera italiana (USI), Bellinzona, Switzerland. [4] YUMAB GmbH, Braunschweig, Germany. [5] Technische Universität Braunschweig, Institut für Genetik, Braunschweig, Germany. [6] BioCopy GmbH, Emmendingen, Germany. [7] Helmholtz Centre for Infection Research, Research Group Innate Immunity and Infection, Braunschweig, Germany. [8] Westfälische Wilhelms-Universität Münster, Institut für Virologie (IVM), Münster, Germany. [9] Centre for Individualised Infection Medicine (CIIM), a joint venture of Helmholtz Centre for Infection Research and Medical School Hannover, Braunschweig, Germany. [10] These authors contributed equally: Federico Bertoglio, Doris Meier, Nora Langreder, Stephan Steinke, Ulfert Rand, Luca Simonelli. [11] These authors jointly supervised this work: Luca Varani, Günter Roth, Luka Čičin-Šain, Maren Schubert, Michael Hust. ✉email: luca.varani@irb.usi.ch; guenter.roth@biocopy.de; Luka.Cicin-Sain@helmholtz-hzi.de; maren.schubert@tu-bs.de; m.hust@tu-bs.de

In 2015, Menachery et al. presciently wrote: "Our work suggests a potential risk of SARS-CoV re-emergence from viruses currently circulating in bat populations"[1]. Four years later, a novel coronavirus causing a severe pneumonia is causing a worldwide pandemic and has been named severe acute respiratory syndrome coronavirus 2 (SARS-CoV-2). The outbreak was initially noticed in a sea food market in Wuhan, Hubei province (China) at the end of 2019. The disease has been named COVID-19 (coronavirus disease 2019) by the World Health Organization. Genome sequencing shows high identity to bat coronaviruses (CoV, in particular RaTG13), beta-CoV virus causing human diseases like SARS and Middle East respiratory syndrome (MERS), and, to a lesser extent, the seasonal CoV hCoV-OC43 and HCov-HKU1[2,3]. The spike (S) protein of SARS-CoV-2, as well as SARS-CoV, binds to the human zinc peptidase angiotensin-converting enzyme 2 (ACE2), which is expressed on numerous cells, including lung, heart, kidney, and intestine cells, thus initiating virus entry into target cells. S protein consists of the N-terminal S1 subunit, which includes the receptor-binding domain (RBD), and the C-terminal S2 subunit, which is anchored to the viral membrane and is required for trimerization of spike itself and fusion of the virus and host membrane[4–6]. The host enzyme furin cleaves the S protein between S1 and S2 during viral formation and the membrane-bound host protease TMPRSS2 is responsible for the proteolytic activation of the S2' site, which is necessary for conformational changes and viral entry[7–10].

Antibodies against the spike protein of coronaviruses are potential candidates for therapeutic development[11]. Antibodies against the S1 subunit, especially against RBD, can potently neutralize SARS-CoV and MERS[12–14]. Monoclonal human antibodies against SARS-CoV are described to cross-react with SARS-CoV-2; some of them are also able to neutralize SARS-CoV-2[15,16]. Other reports show how monoclonal antibodies against SARS-CoV-2 can be selected by rescreening memory B cells from a SARS patient[17], selected from COVID-19 patients by single B cell PCR[18,19] or using phage display[20,21]. Human recombinant antibodies are successfully used for the treatment of other viral diseases. The antibody mAb114[22] and the three antibody cocktail REGN-EB3[23] showed a good efficiency in clinical trials against Ebola virus[24]. The antibody palivizumab is European Medicines Agency (EMA)/Food and Drug Administration (FDA) approved for treatment of a severe respiratory infection of infants caused by the respiratory syncytial virus (RSV)[25,26] and can be used as a guideline to develop therapeutic antibodies against SARS-CoV-2.

Antibody phage display is a powerful tool to generate human antibodies against infectious diseases[27]. We successfully used this technology to develop in vivo protective antibodies against Venezuelan encephalitis virus[28], Western-equine encephalitis[29,30], Marburg virus[31], and Ebola Sudan virus[32].

In this work, we show the generation of human recombinant antibodies against the spike proteins of SARS-CoV-2 from a universal, human naïve antibody gene library that was constructed before the emergence of SARS-CoV-2. Several selected scFv-Fc antibodies efficiently inhibit the binding of the spike protein to ACE2-expressing cells and are blocking SARS-CoV-2 infection of VeroE6 cells. The best antibody in the IgG format is a potential candidate for the clinical development of a passive immunotherapy for therapeutic and prophylactic purposes.

## Results

### SARS CoV2 spike domains or subunits and human ACE2 were produced in insect cells and mammalian cells.
SARS-CoV-2 RBD-SD1 (aa319–591) according to Wrapp et al.[33], S1 subunit (aa14–694), S1-S2 (aa14–1208, with proline substitutions at position 986 and 987 and "GSAS" substitution at the furin site, residues 682–685), and extracellular domain of ACE2 receptor were produced in insect cells using a plasmid-based baculovirus-free system[34] as well as in Expi293F cells. All antigens with exception of S1-S2 were produced with human IgG1 Fc part, murine IgG2a Fc part, or with 6xHis tag in both expression systems. S1-S2 was only produced with 6xHis tag. The extracellular domain of ACE2 was produced with human IgG1 Fc part or mouse IgG2a in Expi293F cells and 6xHis tagged in insect cells. The yields of all the produced proteins are given in Table 1. A graphical overview of all the produced proteins is given in Supplementary Fig. 1. The expressed proteins were analyzed by size exclusion chromatography (SEC; Supplementary Fig. 2).

S1 as well as S1-S2 were more efficiently produced in insect cells compared to Expi293F cells. RBD-SD1 was produced well in both production systems. The binding of the produced spike domains/proteins to ACE2 was validated by enzyme-linked immunosorbent assay (ELISA) and flow cytometric analysis on ACE2-positive cells (Table 1).

### Antibodies were selected by phage display.
Antibodies were selected against SARS-CoV-2 spike S1 subunit in four panning rounds in microtiter plates (MTPs). The following single-clone screening was performed by antigen ELISA in 96-well MTPs, using soluble monoclonal scFv produced in *Escherichia coli*. Subsequently, DNA encoding for the binders was sequenced and unique antibodies were recloned as scFv-Fc fusions.

In detail, three panning strategies were compared. In a first approach (STE70), the lambda (HAL9) and kappa (HAL10)

**Table 1 Antigen production.**

| | High Five cells | | | Expi293F cells | | |
|---|---|---|---|---|---|---|
| | Yield | Binding to ACE2 in ELISA | Binding to ACE2 on cells | Yield | Binding to ACE2 in ELISA | Binding to ACE2 on cells |
| RBD-hFc | 90 mg/L | Yes | Yes | 203 mg/L | Yes | Yes |
| RBD-mFc | 48 mg/L | Yes | Yes | 116 mg/L | Yes | Yes |
| RBD-His | 92 mg/L | Yes | Yes | 35 mg/L | Yes | Yes |
| S1-hFc[a] | 7 mg/L | Yes | Yes | <1 mg/L | No | No |
| S1-hFc | 50 mg/L | Yes | Yes | <1 mg/L | Weak | Yes |
| S1-mFc | 36 mg/L | Yes | Yes | <1 mg/L | Yes | Yes |
| S1-His | 15 mg/L | Yes | Yes | <1 mg/L | Weak | No |
| S1-S2-His | 8 mg/L | Yes | Yes | <1 mg/L | No | No |

Max. production yields of SARS-CoV-2 spike protein/domains in insect cells (High Five) and mammalian cells (Expi293F). Proteins with His-tag produced in High Five cells and S1-hFc were additionally purified by SEC.
[a]With Furin site.

libraries were combined and the antigen S1-hFc (with furin site, produced in High Five cells) was immobilized in phosphate-buffered saline (PBS). Here only seven unique antibodies were identified. In a second approach, the selection was performed separately for HAL10 (STE72) and HAL9 (STE73) using S1-hFc as antigen (with furin site, SEC purified, immobilized in carbonate buffer). Here 90 unique antibodies were selected from HAL10 and 209 from HAL9. In a third approach (STE77 and STE78), S1-hFc produced in Expi293F cells was used (immobilized in carbonate buffer). Here the panning resulted in only three unique antibodies that were not further analyzed in inhibition assays. An overview is given in Table 2.

The antibody subfamily distribution was analyzed and compared to the subfamily distribution in the HAL9/10 library and in vivo (Fig. 1). The phage display-selected antibodies mostly originated from the main gene families VH1 and VH3. Only few antibodies were found using VH4. In 96 of the 309 selected antibodies (31%), the V-gene VH3-23 was used. The V-gene distribution in the lambda light chains was similar to the distribution in the original library. Only antibodies comprising the V-gene VL6-57 were selected from the lambda library HAL10. In antibodies selected from the kappa library, VK2 and VK4 were underrepresented.

**Anti-SARS-CoV-2 scFv-Fc were produced transiently in mammalian cells**. In the interest of rapid throughput to quickly address the growing impact of the COVID-19 pandemic, only a selection of the unique antibodies was chosen for production as scFv-Fc and characterization. Antibodies with potential glycosylation sites in the complementarity determining regions (CDRs), identified by in silico analysis, were excluded. A total of 109 scFv-Fc antibodies were produced in 5 mL culture scale, with yields ranging from of 20 to 440 mg/L.

**Antibodies inhibit the binding of spike to ACE2-positive cells in the scFv-Fc format**. To further select potential therapeutic candidates, an inhibition assay was established using flow cytometry of ACE2-positive cells, measuring competition of S1-S2 trimer binding by scFv-Fc antibodies. The entire spike protein ectodomain was used for this inhibition assay for optimal representation of the viral binding. In a first screening, the 109 scFv-Fc were tested at 1500 nM (molar ratio antibody: S1-S2 30:1). Seventeen antibodies with inhibition better than 75% were selected for further analysis (Fig. 2A, Table 3, and Supplementary Fig. 3). The complete V-gene sequences of these 17 antibodies are given in Supplementary Fig. 4.

To further characterize these 17 antibodies, their inhibition of ACE2 binding was assessed at concentrations from 1500 to 4.7 nM (from 30:1 to ~1:10 Ab:antigen molar ratio) with the same flow cytometric assay (Fig. 2B and Table 3). Antibodies STE72-8E1 and STE73-2E9 showed 50% inhibition of ACE2 binding at a molar ratio of 0.8 antigen-binding sites per spike monomer. For further validation of the direct RBD:ACE2 inhibition, we performed the same assay using a RBD-mFc construct (Fig. 2C). With the exception of two antibodies (STE72-1G5 and STE73-6B10), all antibodies showed high inhibition of binding with molar ratios of 0.3–0.6:1 for STE72-4E12, STE72-8A2, STE72-8A6, STE73-2B2, STE73-2G8, and STE73-9G3.

The inhibition of the 17 antibodies was further validated on human Calu-3 cells, which naturally express ACE2[9] using RBD-mFc (Supplementary Fig. 5A) and S1-S2-His (Supplementary Fig. 5B) showing a stronger inhibition on Calu-3 compared to the transiently overexpressing ACE2-positive Expi293F cells. The Expi293F system allowed an improved estimation of inhibition potency when using the complete S1-S2 spike protein, because the

**Table 2 Antibody selection strategies using the human naïve antibody gene libraries HAL9/10.**

| Antibody selection campaign | Library | Target | Panning rounds | Binders/screened clones | Unique antibodies | Cloned as scFv-Fc | Inhibiting antibodies |
|---|---|---|---|---|---|---|---|
| STE70 | HAL9/10 | S1-hFc (Hi5) | 4 | 7/94 | 7 | 7 | 1 |
| STE72 | HAL10 (kappa) | S1-hFc (Hi5) | 4 | 397/752 | 90 | 44 | 8 |
| STE73 | HAL9 (lambda) | S1-hFc (Hi5) | 4 | 519/846 | 209 | 59 | 8 |
| STE77 | HAL10 (kappa) | S1-hFc (Expi) | 4 | 7/564 | 2 | n.a. | n.a. |
| STE78 | HAL9 (lambda) | S1-hFc (Expi) | 4 | 10/282 | 1 | n.a. | n.a. |

n.a. not analyzed.

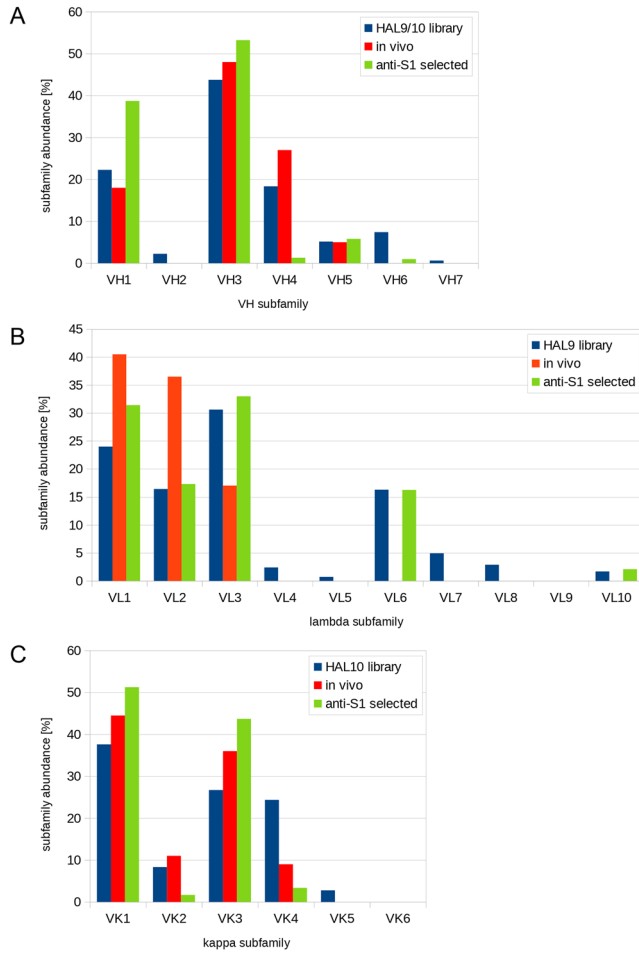

**Fig. 1 Use of V region genes in human anti-SARS-CoV-2 antibodies.**
Comparison of the distribution of V region gene subfamilies in the universal
HAL9/10 library[50], the in vivo distribution of subfamilies[82], and the
distribution of antibodies against S1 selected from HAL9/10. **A** Abundance
of VH, **B** Vκ, and **C** Vλ.

S1-S2 was directly labeled with a fluorophore and the signals were
not amplified in comparison to RBD with a murine Fc and a
fluorophore-labeled secondary antibody. Further, ACE2-expressing
Expi293F cells present a much higher amount of ACE2 receptor on
their surface compared to Calu-3, due to the CMV-mediated
expression (data not shown). Taken together, these data show that
all 17 inhibiting antibodies selected against S1 directly interfered
with RBD-ACE2 binding.

**Determination of EC50 of the inhibiting antibodies to RBD,
S1, and S1-S2.** The EC50 of the inhibiting scFv-Fc on RBD, S1
(without furin site), and S1-S2 spike was measured by ELISA. All
inhibiting antibodies bound the isolated RBD (Fig. 3), identifying
it as their target on the viral surface. Most of the inhibiting
antibodies showed a half-maximal binding in the subnanomolar
range for RBD. While STE72-2G4 showed subnanomolar EC50
values for RBD and S1, it was discarded due to noticeable cross-
reactivity to mFc. The EC50 on the S1-S2 spike trimer was
reduced for most of the antibodies, in comparison to the isolated
RBD or S1.

**ScFv-Fc combinations show synergistic effects in inhibition
assays.** Combinations of best-inhibiting scFv-Fc were tested in
the flow cytometric inhibition assay using 1500 nM antibody
and 50 nM S1-S2 spike (Supplementary Fig. 6). Some of the

combinations showed an increase of inhibition compared to the
same amount of individual antibodies.

**Anti-RBD scFv-Fc neutralize active SARS-CoV-2.** All 17 inhi-
biting scFv-Fc were screened in a cytopathic effect (CPE)-based
neutralization assay using 250 plaque-forming units (pfu)/well
SARS-CoV-2 Münster/FI110320/1/2020 and 1 µg/mL (~10 nM)
scFv-Fc (Fig. 4A and Table 3) to select antibodies for further
characterization as IgG. VeroE6 cells showed pronounced CPE
characterized by rounding and detachment clearly visible in
phase-contrast microscopy upon SARS-CoV-2 infection within
4 days, while uninfected cells maintained an undisturbed con-
fluent monolayer. Virus inoculum preincubated with anti-RBD
antibodies led to decreased CPE in varying degrees quantified by
automated image analysis for cell confluence. The scFv-Fc format
allowed rapid production for neutralization testing and allowed
faster functional pre-screening. All 17 antibodies showed neu-
tralization in this assay. Figure 4B shows examples for strong
(STE73-6C8) and weak (STE73-2C2) neutralizing antibodies and
controls.

**Binding, inhibition, and SARS-CoV-2 neutralization in the
IgG format.** Eleven scFv-Fc showing a neutralization efficacy of
90% according to the CPE-based neutralization assay were con-
verted into the IgG format. First, their binding was analyzed by
titration ELISA on RBD, S1, and S1-S2 (Supplementary Fig. 7 and
Table 3). Three antibodies lost binding after conversion to IgG
(STE72-8A2, STE72-8A6, and STE73-6B10, data not shown),
others showed reduced binding of different degrees, while three
antibodies retained their binding (STE70-1E12, STE72-4E12, and
STE73-2E9). In the next step, the antibodies were tested in the
cell-based inhibition assay using S1-S2 (Fig. 5A) or RBD (Fig. 5B).
Here the inhibition was confirmed for STE73-2E9, -9G3, and
-2G8. STE72-1B6 showed inhibition of RBD comparable to the
latter antibodies, but its activity was almost absent on S1-S2, thus
it was not further considered.

**Inhibiting IgGs are binding at the RBD–ACE2 interface.** The
efficiently inhibiting IgGs STE73-2E9, STE73-2G8, and STE73-
9G3 were analyzed for their binding to various S1 subunit var-
iants harboring a panel of recently reported mutations in RBD
region and the D614G mutation. Three assays were employed:
(1) ELISA (Fig. 6A, B), (2) surface plasmon resonance (SPR)
(Fig. 6B), and (3) bScreen protein array (Fig. 6B) with S1 and/or
RBD-SD1 proteins from different sources. All three antibodies
lost binding to RBD mutations in the region aa483–486 directly at
the RBD–ACE2 interface, showing that mutations in that region
affect their epitope on the antigen. There were only minor dif-
ferences between different approaches, e.g., at positions aa439 and
aa476 for STE73-2E9. We then used this information to guide
and validate computational docking simulations followed by
atomistic molecular dynamics simulations according to protocols
developed and well established in our group[35], obtaining three-
dimensional atomic models of the antibody–RBD interaction for
these three antibodies (Fig. 6C). The binding models of the three
antibodies to the spike ectodomain are shown in Supplementary
Fig. 8.

**STE73-2E9 neutralize SARS-CoV-2 in the IgG format.** As a
final step, STE73-2E9, -9G3, and -2G8 antibodies were analyzed
in a plaque assay using the patient isolate SARS-CoV-2 (Münster/
FI110320/1/2020) to determine their neutralization potency. Here
STE73-2E9 showed an IC50 of 0.43 nM, STE73-9G3 showed only
an IC50 of 1.9 nM, and the IC50 of STE73-2G8 was not deter-
minable as IgG in this assay (Fig. 7A). The IC50 of STE73-2E9

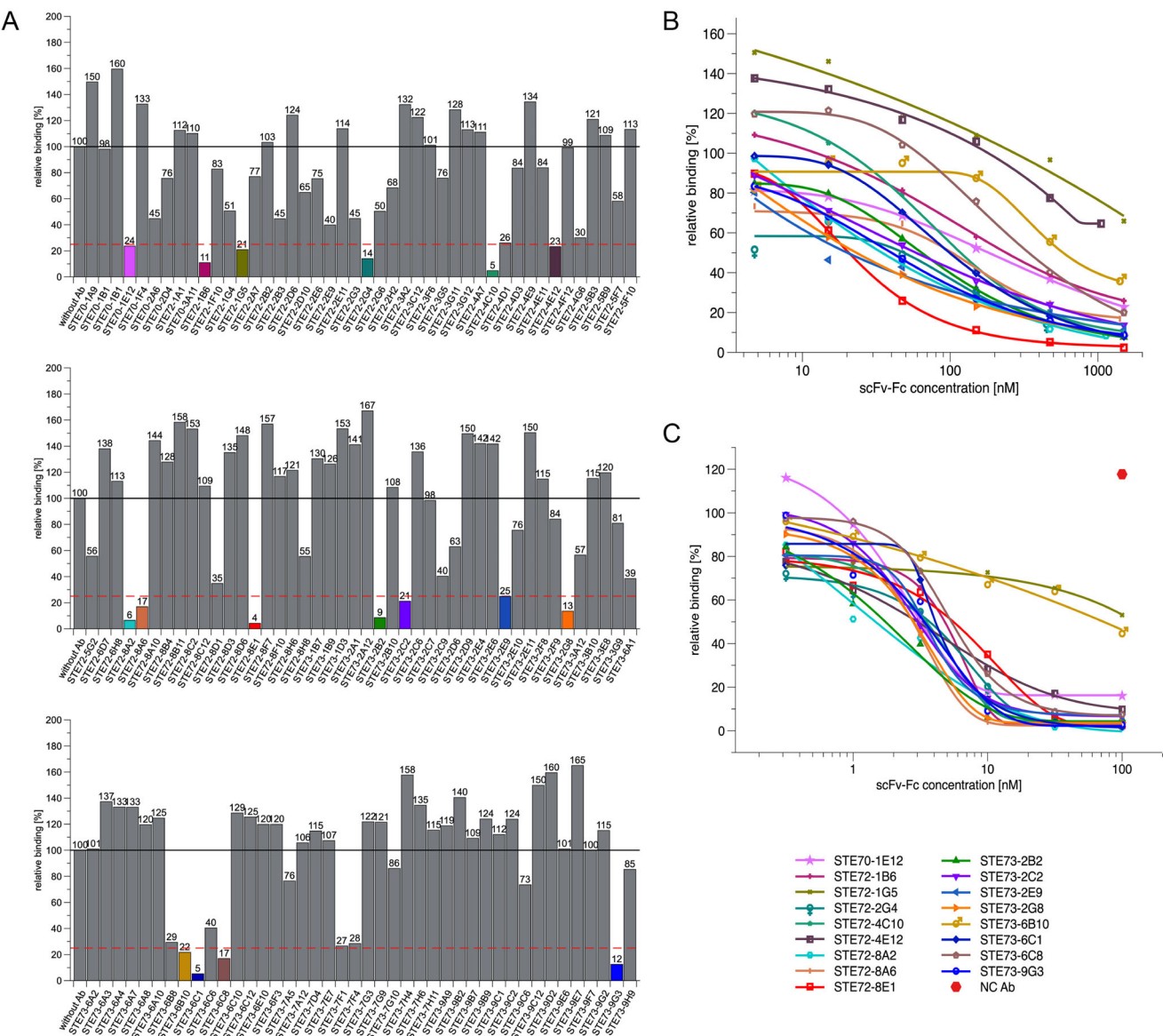

**Fig. 2 Inhibition of SARS-CoV-2 spike protein binding to cell (flow cytometry). A** Inhibition prescreen of 109 scFv-Fc antibodies on ACE2-positive cells using 1500 nM antibody and 50 nM spike protein (30:1 ratio). The antibodies selected for detailed analysis are marked in colors. Data show single measurements. **B** IC50 determination by flow cytometry using 50 nM S1-S2 trimer and 4.7–1500 nM scFv-Fc. **C** IC50 determination by flow cytometry using 10 nM RBD and 0.03–1000 nM scFv-Fc. The inhibition assays were made as single titrations. Logistic5 fit of Origin was used to determine the IC50.

was validated with a second plaque assay with ~150 pfu resulting in an IC50 of 0.41 nM (Fig. 7B).

**Analysis of STE73-2E9 cross-reactivity with other coronaviruses**. The neutralizing antibody STE73-2E9 was further characterized by titration ELISA on SARS-CoV-2 spike recombinant constructs (Fig. 7C) and S1 subunits from different coronaviruses (Fig. 7D) showing that STE73-2E9 is binding specifically SARS-CoV-2 spike protein in the RBD region. The specific binding to S1/RBD of SARS-CoV-2 was further confirmed by the bScreen protein array-binding analysis (data not shown).

**STE73-2E9 binds with nM affinity to RBD**. The affinity of STE73-2E9 was determined by SPR as $2 \times 10^{-9}$ M for RBD-SD1 (Fig. 7E) and $9.25^{-10}$ M for the complete spike protein (Fig. 7F).

**Aggregation behavior of STE73-2E9**. The aggregation behavior of biologicals is a key factor for therapeutic development. STE73-2E9 shows no relevant aggregation under normal conditions (pH 7.4, room temperature (RT) in PBS), heat stress conditions (pH 7.4, 45 °C, 24 h in PBS), and pH stress (pH 3, 24 h, RT), implicating that it has favourable general physicochemical properties that are a prerequisite for the development into a passive vaccine (Supplementary Fig. 9).

**Discussion**

For 130 years, antibodies in animal sera or convalescent human plasma were successfully used for the treatment of infectious diseases, starting with the work of Emil von Behring und Shibasaburo Kitasato against diphtheria[35]. However, the efficacy of human plasma derived from convalescent donors depends on the viral pathogen. In case of Ebola, the survival upon treatment with

**Table 3 Overview on inhibiting antibodies.**

| Antibody name | VH | Germinality index VH [%] | VL | Germinality index VL [%] | EC50 ELISA [nM] scFv-Fc | | | EC50 ELISA [nM] IgG | | | Flow cytometry scFv-Fc spike-binding inhibition assay | | | | scFv-Fc SARS-CoV-2 CPE-based neutralization [%] |
|---|---|---|---|---|---|---|---|---|---|---|---|---|---|---|---|
| | | | | | RBD | S1 | S1-S2 | RBD | S1 | S1-S2 | IC50 [nM] with 50 nM spike | Molar ratio antibody arm: spike | IC50 [nM] with 10 nM RBD | Molar ratio antibody arm: RBD | |
| STE70-1E12 | VH1-2 | 96.7 | VL6-57 | 94.4 | 0.3 | 0.5 | 1.1 | 0.3 | 0.4 | 0.8 | 180 | 7.2 | 3.2 | 0.64 | 98 |
| STE72-1B6 | VH3-23 | 93.4 | VK1-12 | 95.5 | 0.5 | 0.7 | 1.4 | 1.1 | 1.6 | 2.4 | 240 | 9.6 | 4.8 | 0.96 | 90 |
| STE72-1G5 | VH1-69 | 98.9 | VK3-20 | 96.6 | 2.8 | 3.4 | 5.2 | n.a. | | | n.d. | n.d. | n.d. | n.d. | 77 |
| STE72-4C10 | VH3-30 | 97.8 | VK1D-39 | 92.1 | 0.5 | 1 | 2.4 | n.a. | | | 117 | 4.8 | 3.5 | 0.7 | 87 |
| STE72-4E12 | VH1-46 | 100 | VK3-15 | 98.9 | 1.5 | 3.3 | 3.7 | 1.6 | 1.1 | 1.7 | n.d. | n.d. | 3.0 | 0.6 | 99 |
| STE72-8A2 | VH1-18 | 100 | VK1D-33 | 97.8 | 0.5 | 0.7 | 0.8 | Not binding as IgG | | | 35 | 1.4 | 1.5 | 0.3 | 97 |
| STE72-8A6 | VH1-18 | 100 | VK1-5 | 94.4 | 0.5 | 0.9 | 1.2 | Not binding as IgG | | | 102 | 4.0 | 2.8 | 0.56 | 100 |
| STE72-8E1 | VH4-61 | 93.4 | VK1-5 | 93.3 | 0.4 | 0.8 | 0.6 | n.a. | | | 20 | 0.8 | 5.6 | 1.1 | 85 |
| STE72-2G4 | VH1-2 | 100 | VL2-8 | 94.3 | 0.2 | 0.3 | 0.3 | n.a. | | | 37 | 1.4 | 3.7 | 0.74 | 86 |
| STE73-2B2 | VH1-2 | 95.6 | VL6-57 | 92.2 | 0.2 | 0.3 | 0.3 | n.a. | | | 63 | 2.6 | 1.7 | 0.4 | 75 |
| STE73-2C2 | VH3-66 | 96.7 | VL6-57 | 92.2 | 3.1 | 5.7 | 7.8 | n.a. | | | 59 | 2.4 | 3.0 | 0.6 | 70 |
| STE73-2E9 | VH1-18 | 100 | VL1-36 | 96.6 | 0.2 | 0.2 | 0.2 | 0.2 | 0.2 | 0.2 | 20 | 0.8 | 3.4 | 0.68 | 90 |
| STE73-2G8 | VH3-66 | 92.3 | VL3-19 | 100 | 0.2 | 0.3 | 0.4 | 0.8 | 1.7 | 2.8 | 23 | 1.0 | 2.8 | 0.56 | 98 |
| STE73-6B10 | VH1-2 | 97.8 | VL2-11 | 94.3 | 5.5 | 4.9 | 20.2 | Not binding as IgG | | | 612 | 24 | 73 | 14.6 | 90 |
| STE73-6C1 | VH3-30 | 98.9 | VL1-40 | 92.0 | 0.6 | 0.9 | 1.8 | 14.1 | 19.5 | 34 | 97 | 3.8 | 4.1 | 0.81 | 100 |
| STE73-6C8 | VH1-69 | 98.9 | VL6-57 | 93.3 | 1.1 | 1.9 | 5.4 | 2.9 | 4.3 | 4 | 332 | 13.2 | 5.4 | 1.08 | 100 |
| STE73-9G3 | VH3-23 | 97.8 | VL1-40 | 94.3 | 0.4 | 0.5 | 0.9 | 0.4 | 0.9 | 2.3 | 40 | 1.6 | 3.4 | 0.6 | 100 |

V-genes were determined by VBASE2 (vbase2.org)[72]. The EC50 were measured on 30 ng immobilized RBD-mFc, S1-mFc, S1-S2-His (trimer) by ELISA. The IC50 was measured by flow cytometry using 50 nM (in relation to monomer) S1-S2 trimer, respectively, 10 nM RBD, and ACE2-positive cells. The molar ratio of antibody-binding site: S1-S2 or RBD is given for 50% inhibition. CPE-based neutralization assay was performed with 250 pfu/well SARS-CoV-2 and 1 µg/mL (~100 nM) scFv-Fc (median neutralization %). EC50 were calculated with GraphPad Prism Version 6.1, fitting to a four-parameter logistic curve. IC50 values were calculated using Logistic5 fit of Origin.
*n.a.* not applicable.

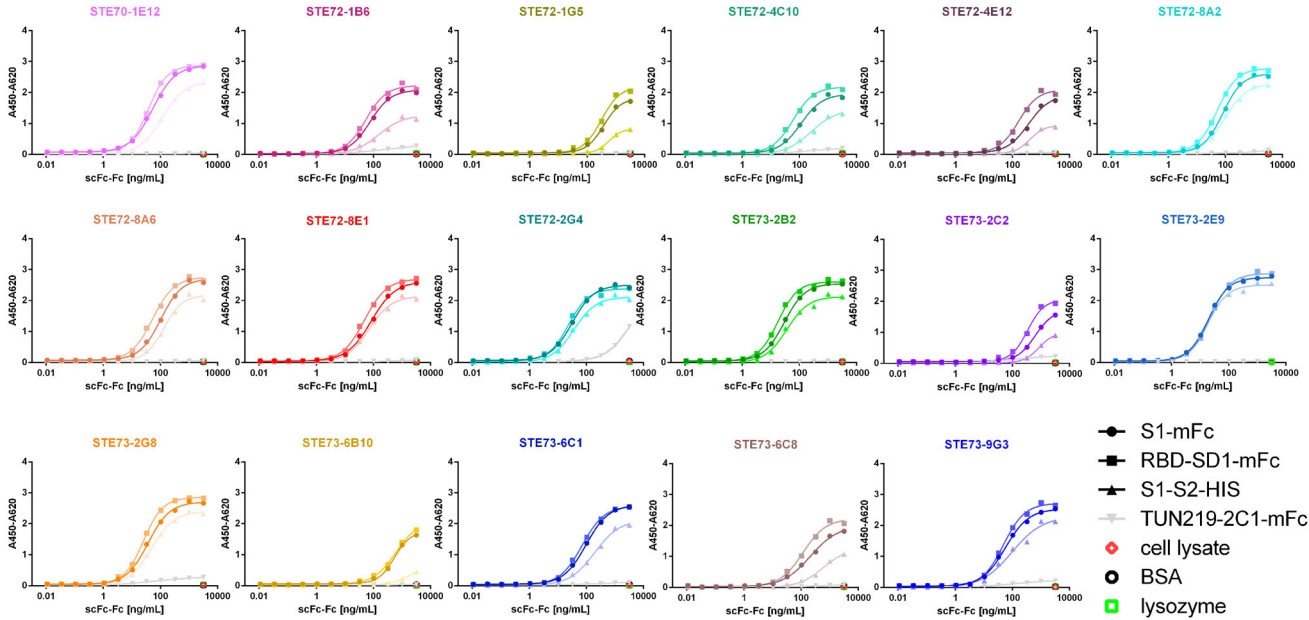

**Fig. 3 Determination of EC50 on RBD.** Binding in titration ELISA of the 17 best inhibiting scFv-Fc on RBD (fusion protein with murine Fc part), S1 (fusion protein with murine Fc part), or S1-S2 (fusion protein with His tag). Sequence SARS-CoV-2 (Gene bank QHD43416). An unrelated antibody with murine Fc part (TUN219-2C1), human HEK293 cell lysate, BSA, or lysozyme were used as controls. Experiments were performed in duplicate and mean values are given. EC50 were calculated with GraphPad Prism Version 6.1, fitting to a four-parameter logistic curve.

convalescent human plasma was not significantly improved over the control group[36]. On the other hand, reduced mortality and safety was shown for convalescent plasma transfer in case of influenza A H1N1 in 2009[37,38]. This approach was also used against emerging coronaviruses. While the outcomes were not significantly improved in a very limited number of MERS patients[39], the treatment was successful for SARS[40,41]. This approach was also used for COVID-19 with promising results[42]. The mode of action of these polyclonal antibody preparations may vary, including virus neutralization, Fcγ receptor-binding-mediated phagocytosis, or antibody-dependent cellular cytotoxicity (ADCC) as well as complement activation[43–45]. In any serum therapy, the composition and efficacy of convalescent plasma is expected to differ from donor to donor, as well as from batch to batch, and sera must be carefully controlled for viral contaminations (e.g., HIV, hepatitis viruses) and neutralization potency. A convalescent patient can provide 400–800 mL plasma, with 250–300 mL of plasma typically needed per treatment. With two rounds of treatment per patient, this is a grave limitation, since one donor can only provide material for 1–2 patients[42,43]. Human or humanized monoclonal antibodies are a powerful alternative to polyclonal antibodies derived from convalescent plasma. Following this approach, the humanized antibody Palivizumab was approved in 2009 for treatment and prevention of RSV infections[46]. Other antibodies against viral diseases successfully tested in clinical studies are mAb114 and REGN-EB3 against Ebola disease[24].

Phage display-derived antibodies are typically well-established medications: 12 such antibodies are approved by EMA/FDA at the time of writing, a significant increase compared to the six such antibodies approved in 2016[47]. In this work, we used phage display to isolate monoclonal human antibodies capable of neutralizing SARS-CoV-2 from a universal, naïve antibody gene library that was generated from healthy donors before the emergence of the SARS-CoV-2 virus. This allowed selection of human antibodies against this virus without the necessity to obtain material from COVID-19-infected individuals. While most antibodies against SARS-CoV-2 were obtained from convalescent

patients with few exceptions[20,21,48,49], our approach demonstrates that human antibodies can be generated without the necessity to wait for material from COVID-19-infected individuals. With SARS-CoV-2, the whole world invested an unprecedented amount of resources toward a single goal. This is not common for other diseases and infective outbreaks. The fact that research groups and companies were able to rapidly recruit many individuals and quickly find potentially good antibodies should not be held as standard for every situation. Therefore, this strategy offers a very fast additional opportunity to respond to future pandemics.

As the human receptor of SARS-CoV and SARS-CoV-2 is ACE2[3], we focused on antibodies which directly block the interaction of the spike protein with this receptor and antibodies preventing ACE2 binding were shown to potently neutralize the closely related SARS-CoV virus[16]. Three hundred and nine unique fully human monoclonal antibodies were generated using different panning strategies. The S1 subunit produced in insect cells was better suited for antibody selection than the S1 subunit produced in mammalian cells. The V-gene distribution of the selected anti-Spike antibodies is largely in accordance with the V-gene subfamily distribution shown by Kügler et al.[50] for antibodies selected against 121 other antigens from HAL9/10. Only the VH1 subfamily was overrepresented and VH4 and Vkappa4 subfamilies were rarely selected despite their presence in the HAL libraries. The most frequently used V-gene was VH3-30. Interestingly, an increased use of this V-gene in anti-SARS-CoV-2 antibodies was also described by Robbiani et al.[51] for anti-RBD B cells selected from COVID-19 patients. By contrast, the second most selected V-gene was VH3-53, which was selected in our approach only once. Robbiani et al. also described an overrepresented use of VL6-57, as found in our antibodies as well. However, it has to be noted that VL6 is also overrepresented in our naïve library compared to its in vivo occurrence.

From the initial 309 scFv, 109 were recloned in the scFv-Fc IgG-like bivalent format. Their ability to inhibit binding of fluorescently labeled S1-S2 trimer to ACE2-expressing cells was assessed by flow cytometry. The half-maximal inhibition of the best inhibiting 17 scFv-Fc was measured both with the spike

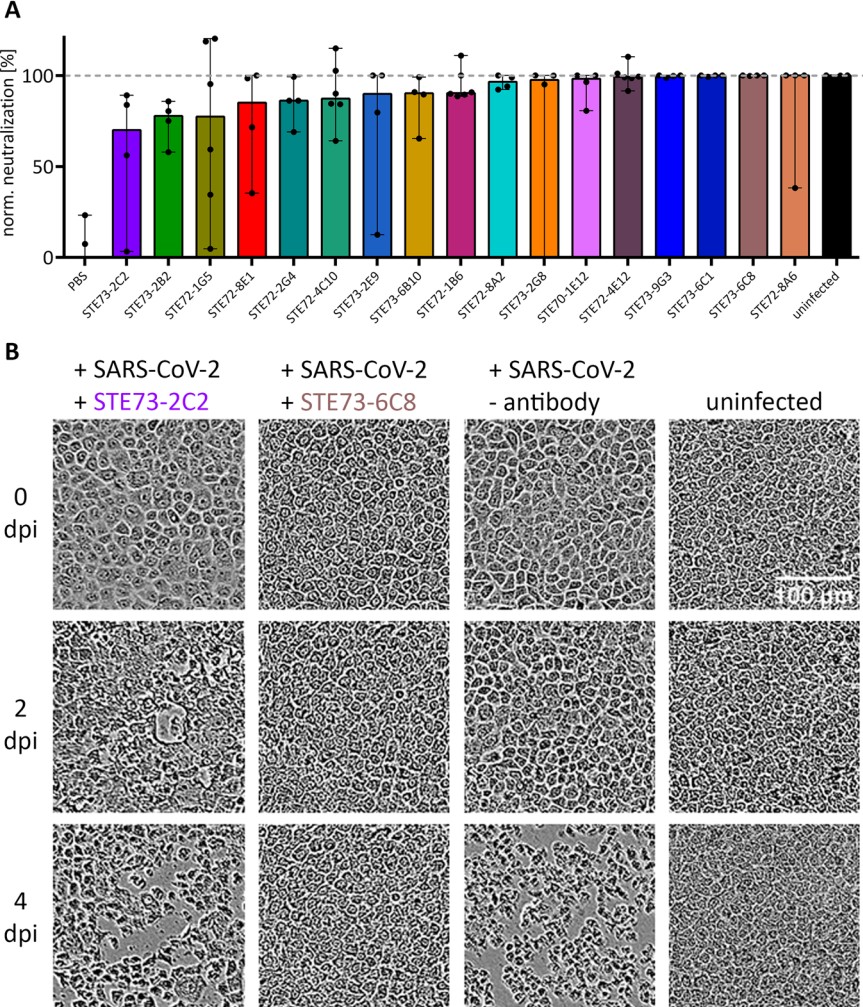

**Fig. 4 SARS-CoV-2 neutralization in the scFv-Fc format.** Neutralization analysis using 250 pfu of SARS-CoV-2 in a CPE-based neutralization assay. **A** Cell monolayer occupancy at 4 days post infection in the absence of neutralizing antibodies was compared to uninfected control cells and median values were normalized as 0 and 100% occupancy, respectively. Histograms indicate medians of normalized monolayer occupancy in a neutralization assay using 1 μg/mL (~10 nM) antibody for each of the 17 tested antibodies. Data show the median from 4 or 6 replicates, the black dots indicate monolayer occupancy in individual assays, and the range is given for the maximum and minimum measurements. **B** Representative phase-contrast microscopic pictures of uninfected cells, cells infected in the absence of antibodies, in the presence of a poorly neutralizing scFv-Fc (STE73-2C2), or of a highly neutralizing scFv-Fc (STE73-6C8).

trimer and isolated RBD. Significantly, some of the antibodies showed half-maximal inhibition at a ratio around 1:1—in certain cases even better—when calculated per individual binding site (antigen-binding site:spike monomer/RBD). A similar molar ratio of 1:1 was demonstrated by Miethe et al. for inhibition of botulinum toxin A[52]. In the trimeric spike protein, the RBD can be in an "up" (open) or "down" (close) position. The "down" conformation cannot bind to ACE2, in contrast to the less stable "up" conformation[33]. The RBDs can be in different conformations on the same spike trimer, which offers a possible explanation for the observed effective antibody to spike molar ratios lower than 1:1. This is in accordance with the cryo-electron microscopic images recorded by Walls et al.[4], where they could find half of the recorded trimers with one RBD in the open conformation. We observed that molar ratios for half maximal inhibition were lower for RBD compared to spike protein. For some antibodies, approximately 0.5 antigen-binding sites were needed to achieve a 50% inhibition. The fact that the antibodies are more efficient at inhibiting RBD binding to ACE2 rather than S1-S2 trimer binding can be explained with the higher affinity of the antibodies

for the isolated RBD compared to the trimeric spike, which in turn points to the presence of partially or completely inaccessible epitopes on the trimer, an occurrence seen in other viruses. This is similar to what was reported by Pinto et al.[17], who also showed a lower affinity of the antibody S309 for spike compared to RBD.

Inhibition of ACE2 binding was stronger on the human lung cells Calu-3, which better represent the in vivo situation than transiently ACE2-overexpressing cells. Nevertheless, we did the titration assays on ACE2-overexpressing Expi293F cells because these seemed to allow a better quantitative discrimination of inhibiting potency.

Antibody combinations can have a synergistic effect as previously described for toxins and viruses[32,53–55]. This approach may also avoid formation of viral escape mutants. Here the best combinations showed a significantly improved inhibition efficacy, at least when using an excess of antibodies (Ab:Ag molar ratio 30:1).

All of the 17 scFv-Fc were tested in neutralization assays using a SARS-CoV-2 strain isolated from a patient and all antibodies showed a degree of neutralization in this assay. While this study

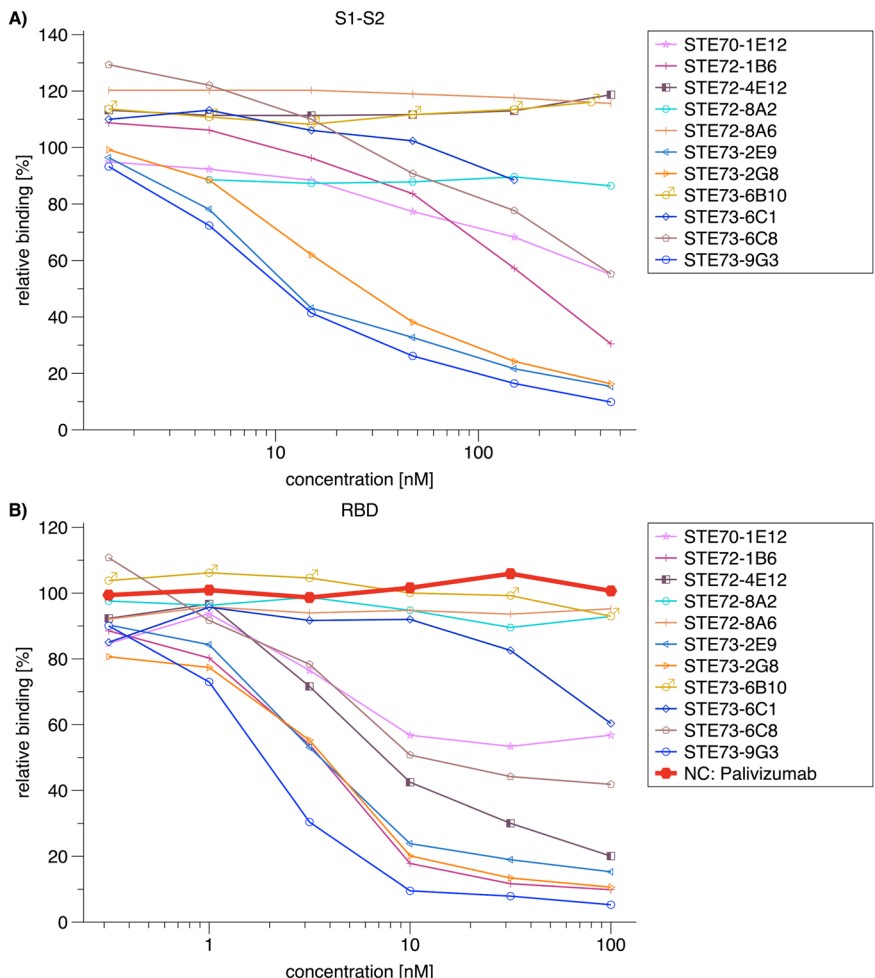

**Fig. 5 Inhibition of RBD–ACE2 interaction by IgG. A** IC50 determination by flow cytometry using 50 nM S1-S2-His and 0.5-500 nM IgG. **B** IC50 determination by flow cytometry using 10 nM RBD-mFC and 0.1–100 nM IgG. Palivizumab was used as negative control. The inhibition assays were made as single titrations. Logistic5 fit of Origin was used to determine the IC50.

did not aim to define the lowest effective concentration of individual antibodies in limiting dilution conditions, all tested antibodies showed a clear and measurable effect at a relatively low concentration. Therefore, our approach provided a rapid selection of antiviral antibodies.

In a next step, we converted 11 antibodies with the best neutralization efficacy according to the cytopathic assay into the IgG format. It was completely unexpected that most antibodies lost efficacy in the inhibition assay after conversion from scFv-Fc to IgG including antibodies like STE70-1E12 without loss of affinity according to the titration ELISA. These results are in contrast to former results where none[54] or only a low percentage[55,56] of antibodies lost efficacy after conversion from scFv-Fc to IgG. Nevertheless, three antibodies showed a good inhibition in the cell-based assay and did not bind to the region of aa483–486 known for RBD mutations from publications[19,57] and from GISAID database (www.gisaid.org). Experimentally validated computational docking shows that these antibodies still at least partially occupy the ACE2-binding site on the RBD, thus likely achieving direct inhibition of virus–ACE2 interaction. The binding sites of antibody BD368-2[18], B38[58], and REGN10933[59] also overlap with the RBD–ACE2 binding interface. The best neutralizing antibody STE73-2E9 showed an IC50 of 0.41 nM, which is higher compared to antibodies derived from COVID-19 patients. Cao et al.[18] reported an IC50 of 33 ng/mL (~0.22 nM)

for BD368-2 in a comparable live virus plaque assay. Other publications reports better IC50 efficacies, e.g., Kreye et al.[60] for CV07-209 with 3.1 ng/mL (~0.02 nM) or Rogers et al.[61] for CC6.33 with 1 ng/mL. Some assays are often not directly comparable, e.g. Rogers et al. used SARS-CoV-2 pseudovirus instead of authentic virus.

The neutralizing antibody STE73-2E9 was specific for SARS-CoV-2, and we conclude that this antibody is a suitable candidate for the development of passive immunotherapy for the treatment of COVID-19. It could be used not only therapeutically to prevent individuals from being hospitalized in intensive care units but also prophylactically to protect health care workers or risk groups who do not respond to vaccination. Before clinical application, the risk of antibody-dependent enhancement (ADE) of disease has to be considered for COVID-19. In contrast to antibodies against Ebola where ADCC is important for protection[22], antibodies directed against the spike protein of SARS-CoV-2 may lead to ADE[62–64]. SARS cause an acute lung injury, which is also driven by immune dysregulation and inflammation caused by anti-spike antibodies[65]. While Quinlan et al.[66] described that animals immunized with RBD SARS-CoV-2 did not mediate ADE and suggested for vaccines the use of RBD, some of the monoclonal antibodies we analyzed in this study lead to an increased binding of the spike protein to ACE2-positive cells. A possible explanation could be multimerization of the spike by

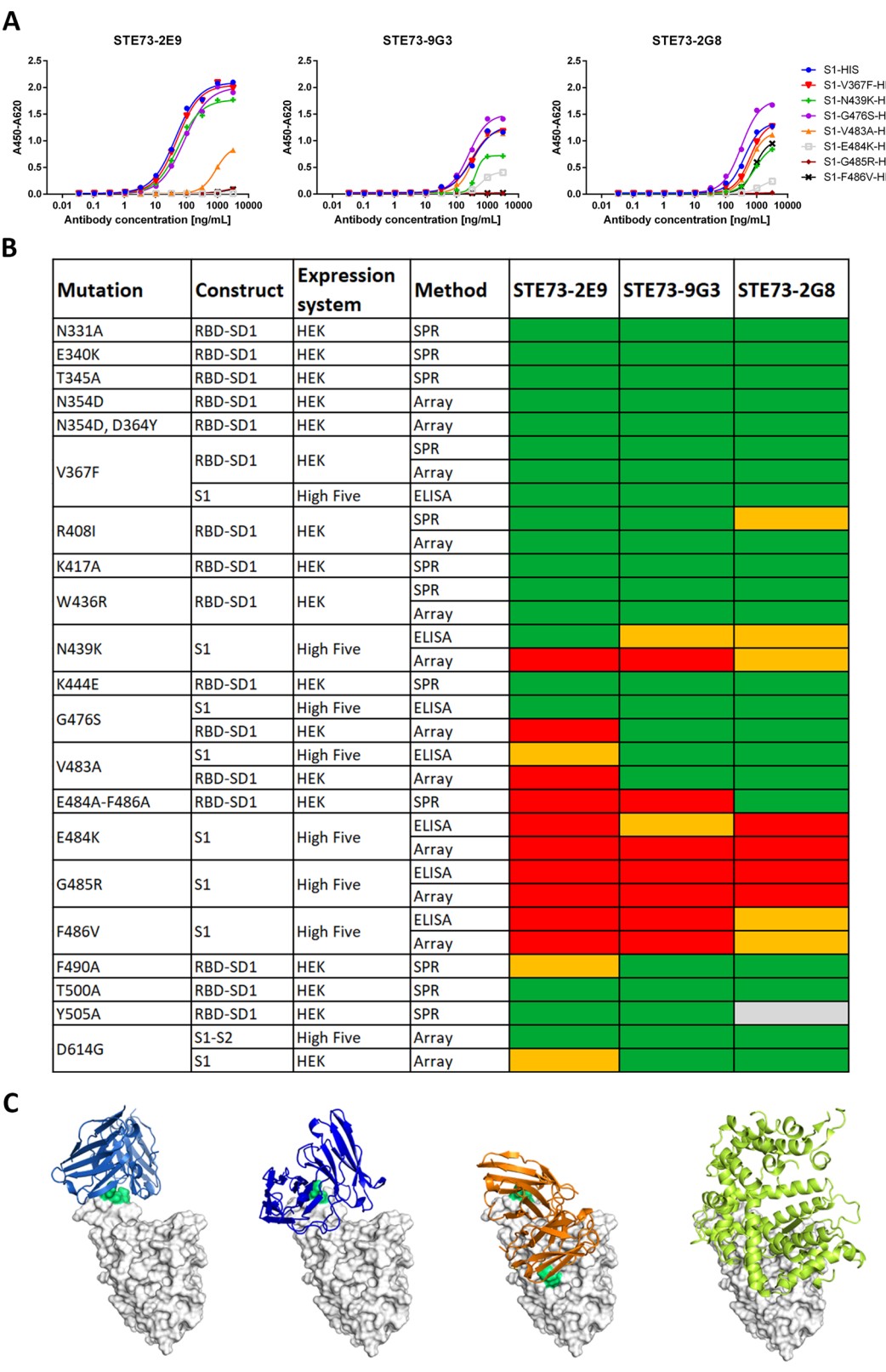

**Fig. 6 Binding to RBD mutants, epitopes, and structure models. A** ELISA using STE73-2E9, -9G3, and -2G8 on S1-His with different RBD mutations. **B** Overview of the binding of STE73-2E9, -9G3, and -2G8 to different RBD mutations analyzed by ELISA, SPR, and protein array. Sequence SARS-CoV-2 (Gene bank QHD43416). ELISA experiments were performed in duplicate and mean values are given. **C** The three antibodies STE73-2E9, -9G3, and -2G8 are binding to the ACE–RBD interface (docking models based on epitope data from binding to RBD mutations). Experimentally validated computational models of the variable regions of the antibodies (colored cartoons) binding to the RBD (white surface, same orientation in all images) are shown. The cartoon representation of ACE2 is also shown for comparison.

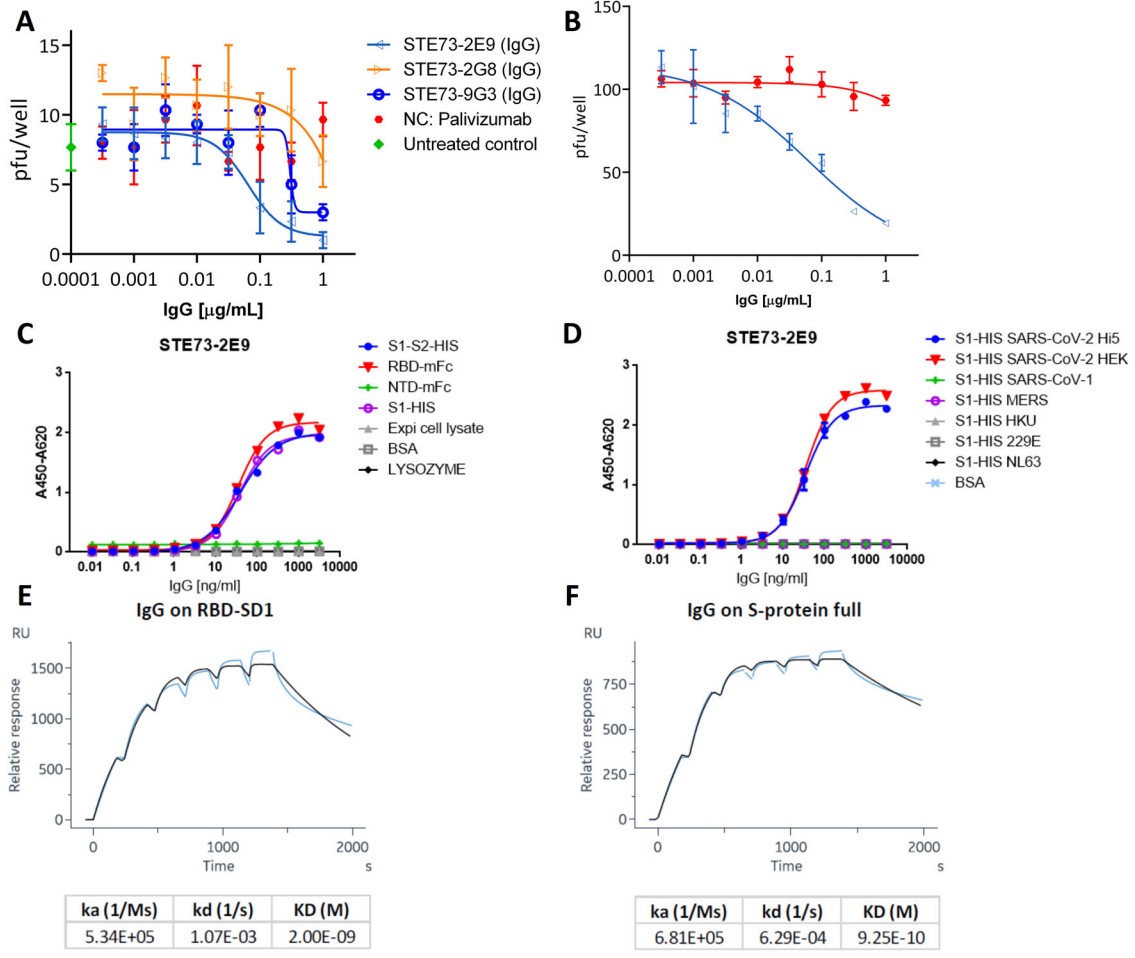

**Fig. 7 Characterization of the neutralizing antibody STE73-2E9 in IgG format. A** Neutralization of 20–30 pfu SARS-CoV-2 by STE73-2E9, -9G3, and -2G8. Palivizumab was used as isotype control. **B** Validation of neutralization potency of STE73-2E9 using 100 pfu. Neutralization assays were performed in triplicates, mean ± s.e.m. are given. **C** Titration ELISA on the indicated antigens. ELISA shows single titration of two representative experiments (see also Supplementary Fig. 7). **D** Cross-reactivity to other coronavirus spike proteins analzyed by ELISA. S1-HIS SARS-CoV-2 Hi5 was produced in house. S1-HIS SARS-CoV-2 HEK and all other coronavirus S1 domain proteins were obtained commercially. ELISA experiments were performed in duplicate and the mean values are given. **E, F** Kinetic parameter determination through single-cycle kinetic titration SPR of STE73-2E9 IgG on HEK cell produced RBD-SD1 and S1-S2, respectively (concentrations: 200, 100, 50, 25, 12.5, 6.25 nM).

antibody "cross-linking" in this assay or the stabilization of an infection-promoting conformation by the antibodies. These aspects need to be carefully considered in any development of therapeutic antibodies; we suggest to focus on RBD and/or the use of silenced Fc parts with deleted Fcγ and C1q binding[67–69] for safety reasons.

In conclusion, we report the successful isolation and characterization of a fully human, recombinant anti-spike neutralizing monoclonal antibody from naïve phage display libraries. Our approach demonstrates how neutralizing antibodies can be efficiently selected in a rapid time frame and without the need of convalescent patient material. Furthermore, the strategy we used efficiently targeted the spike:ACE interface allowing the selection of directly blocking antibodies.

## Methods

**Design of expression vectors.** Production in Expi293F cells was performed using pCSE2.5-His-XP, pCSE2.6-hFc-XP, or pCSE2.6-mFc-XP[70] where the respective single-chain variable fragment of the antibodies or antigens were inserted by *Nco*I/*Not*I (NEB Biolabs) digestion. Antigen production in High Five insect cells was performed using *Nco*I/*Not*I compatible variants of the OpiE2 plasmid[71] containing an N-terminal signal peptide of the mouse Ig heavy chain; the respective antigen; and C-terminal 6xHis-tag, hFc, or mFc. Single point mutations in S1-HIS constructs were inserted through site-directed mutagenesis using overlapping primers

according to Zheng et al.[72] with slight modifications: S7 fusion polymerase (Mobidiag, Espoo, Finland) with the provided GC buffer and 3% dimethyl sulfoxide was used for the amplification reaction. The used olignucleotide primers for cloning and site-directed mutagenesis are given in Supplementary Table 1.

**Production of antigens in insect cells.** Different domains or subunits of the spike protein (GenBank: MN908947), S1 subunit mutants, and the extracellular domain of ACE2 receptor (GenBank NM_021804.3) were baculovirus-free produced in High Five cells (Thermo Fisher Scientific) by transient transfection as previously described in Bleckmann et al.[34]. Briefly, High Five cells were cultivated at 27 °C, 110 rpm in ExCell405 media (Sigma) and kept at a cell density between $0.3 \times 10^6$ and $5.5 \times 10^6$ cells/mL. For transfection, cells were centrifuged and resuspended in fresh media to a density of $4 \times 10^6$ cells/mL and transfected with 4 µg plasmid/mL and 16 µg/mL of PEI 40 kDa (Polysciences). Four hours up to 24 h after transfection, cells were fed with 75% of the transfection volume. At 48 h after transfection, cell culture medium was doubled. Cell supernatant was harvested 5 days after transfection in a two-step centrifugation (4 min at $180 \times g$ and 20 min at above $3500 \times g$) and 0.2 µm filtered for purification.

**Production of antigens and scFv-Fc in mammalian cells.** Antibodies, different domains, or subunits of the spike protein and the extracellular domain of ACE2 were produced in Expi293F cells (Thermo Fisher Scientific). Expi293F cells were cultured at 37 °C, 110 rpm, and 5% $CO_2$ in Gibco FreeStyle F17 expression media (Thermo Fisher Scientific) supplemented with 8 mM Glutamine and 0.1% Pluronic F68 (PAN Biotech). At the day of transfection, cell density was between $1.5 \times 10^6$ and $2 \times 10^6$ cells/mL and viability at least >90%. For formation of DNA:PEI complexes, 1 µg DNA/mL transfection volume and 5 µg of 40 kDa PEI

(Polysciences) were first diluted separately in 5% transfection volume in supplemented F17 media. DNA and PEI was then mixed and incubated ~25 min at RT before addition to the cells. Forty-eight hours later, the culture volume was doubled by feeding HyClone SFM4Transfx-293 media (GE Healthcare) supplemented with 8 mM Glutamine. Additionally, HyClone Boost 6 supplement (GE Healthcare) was added with 10% of the end volume. One week after transfection, supernatant was harvested by 15-min centrifugation at $1500 \times g$.

**Protein purification**. Protein purification was performed depending on the production scale in either 24-well filter plate with 0.5 mL resin (10 mL scale) or 1 mL column on Äkta go (Cytiva), Äkta Pure (Cytiva), or Profina System (BIO-RAD). MabSelect SuRe or HiTrap Fibro PrismA (Cytiva) was used as resins for Protein A purification. For His-tag purification of Expi293F, supernatant HisTrap FF Crude column (Cytiva) and for His-tag purification of insect cell supernatant HisTrap excel column (Cytiva) was used. All purifications were performed according to the manufacturer's manual. Indicated antigens were further purified by SEC by a 16/600 Superdex 200 kDa pg (Cytiva). All antigens, antibodies, and scFv-Fc were run on Superdex 200 Increase 10/300GL (Cytiva) on Äkta or HPLC (Techlab) on an AdvanceBio SEC 300 Å 2.7 µm, 7.8 × 300 mm (Agilent) for quality control.

**Validation of spike protein binding to ACE2**. ACE2 binding to the produced antigens was confirmed in ELISA and on cells in flow cytometry. For ELISA, 200 ng ACE2-mFc per well was immobilized on a Costar High binding 96-well plate (Corning, Costar) at 4 °C overnight. Next, the wells were blocked with 350 µL 2% MBPST (2% (w/v) milk powder in PBS; 0.05% Tween20) for 1 h at RT and then washed 3 times with $H_2O$ and 0.05% Tween20 (BioTek Instruments, EL405). Afterwards, the respective antigen was added at the indicated concentrations and incubated 1 h at RT prior to another 3 times washing step. Finally, the antigen was detected by mouse-anti-polyHis conjugated with horseradish peroxidase (HRP) (1:20,000, A7058, Sigma) for His-tagged antigens, goat-anti-mIgG(Fc) conjugated with HRP (1:42,000, A0168, Sigma) for mFc tagged antigen versions, or goat-anti-hIgG(Fc) conjugated with HRP (1:70,000, A0170, Sigma) if hFc-tagged antigens had to be detected. Bound antigens were visualized with tetramethylbenzidine (TMB) substrate (20 parts TMB solution A (30 mM potassium citrate; 1 % (w/v) citric acid (pH 4.1)) and 1 part TMB solution B (10 mM TMB; 10% (v/v) acetone; 90% (v/v) ethanol; 80 mM $H_2O_2$ (30%)) were mixed). After addition of 1 N $H_2SO_4$ to stop the reaction, absorbance at 450 nm with a 620-nm reference wavelength was measured in an ELISA plate reader (BioTek Instruments, Epoch).

To verify the ACE2–antigen interaction on living cells, Expi293F cells were transfected according to the protocol above using pCSE2.5-ACE2$_{fl}$-His and 5% enhanced green fluorescent protein (GFP) plasmid. The gating strategy is shown in Supplementary Fig. 3. Two days after transfection, purified S1-S2-His, S1-His, or RBD-His were labeled using the Monolith NT$^{TM}$ His-Tag Labeling Kit RED-tris-NTA (Nanotemper) according to the manufacturer's protocol. Fc-tagged ligand versions were labeled indirectly by using goat-anti-mFc-APC (1:50, 115-136-071, Dianova) or mouse anti-hFcγ-APC (1:50, Clone HP6017, Biolegend) antibody. In all, 100, 50, and 25 nM of antigen were incubated with $5 \times 10^5$ ACE2-expressing or non-transfected Expi293F cells (negative control) 50 min on ice. After two washing steps, fluorescence was measured in MACSQuant Analyzer (Miltenyi Biotec).

**Antibody selection using phage display**. The antibody selection was performed as described previously with modifications[73]. In brief, for panning procedure, the antigen was immobilized on a High binding 96-well plate (Corning, Costar). Five micrograms of S1-hFc (produced in High Five cells) was diluted in carbonate buffer (50 mM NaHCO₃/Na₂CO₃, pH 9.6) and coated onto the wells at 4 °C overnight. Next, the wells were blocked with 350 µL 2% MBPST (2% (w/v) milk powder in PBS; 0.05% Tween20) for 1 h at RT and then washed 3 times with PBST (PBS; 0.05% Tween20). Before adding the libraries to the coated wells, the libraries ($5 \times 10^{10}$ phage particles) were preincubated with 5 µg of an unrelated scFv-Fc and 2% MPBST on blocked wells for 1 h at RT, to deprive libraries of human Fc fragment binders. The libraries were transferred to the antigen-coated wells, incubated for 2 h at RT, and washed 10 times. Bound phage were eluted with 150 µL trypsin (10 µg/mL) at 37 °C, 30 min and used for the next panning round. The eluted phage solution was transferred to a 96 deep well plate (Greiner Bio-One, Frickenhausen, Germany) and incubated with 150 µL E. coli TG1 (OD₆₀₀ = 0.5) first for 30 min at 37 °C, then 30 min at 37 °C and 650 rpm to infect the phage particles. One milliliter 2xYT-GA (1.6% (w/v) Tryptone; 1 % (w/v) Yeast extract; 0.5% (w/v) NaCl (pH 7.0), 100 mM D-Glucose, 100 µg/mL ampicillin) was added and incubated for 1 h at 37 °C and 650 rpm, followed by addition of $1 \times 10^{10}$ colony-forming units M13KO7 helper phage. Subsequently, the infected bacteria were incubated for 30 min at 37 °C followed by 30 min at 37 °C and 650 rpm before centrifugation for 10 min at $3220 \times g$. The supernatant was discarded and the pellet resuspended in fresh 2xYT-AK (1.6% (w/v) Tryptone; 1 % (w/v) Yeast extract; 0.5% (w/v) NaCl (pH 7.0), 100 µg/mL ampicillin, 50 µg/mL kanamycin). The antibody phage were amplified overnight at 30 °C and 650 rpm and used for the next panning round. In total, four panning rounds were performed. In each round, the stringency of the washing procedure was increased (20× in panning round 2, 30× in panning round 3, 40× in panning round 4) and the amount of antigen was reduced (2.5 µg in panning round 2, 1.5 µg in panning round 3, and 1 µg in panning round 4). After

the fourth as well as third panning round, single clones containing plates were used to select monoclonal antibody clones for the screening ELISA.

**Screening of monoclonal recombinant binders using E. coli scFv supernatant**. Soluble antibody fragments (scFv) were produced in 96-well polypropylene MTPs (U96 PP, Greiner Bio-One) as described before[55,73]. Briefly, 150 µL 2xYT-GA was inoculated with the bacteria bearing scFv-expressing phagemids. MTPs were incubated overnight at 37 °C and 800 rpm in a MTP shaker (Thermoshaker PST-60HL-4, Lab4You, Berlin, Germany). A volume of 180 µL 2xYT-GA in a PP-MTP well was inoculated with 20 µL of the overnight culture and grown at 37 °C and 800 rpm for 90 min (approx. OD₆₀₀ of 0.5). Bacteria were harvested by centrifugation for 10 min at $3220 \times g$, and the supernatant was discarded. To induce expression of the antibody genes, the pellets were resuspended in 200 µL 2xYT supplemented with 100 µg/mL ampicillin and 50 µM isopropyl-beta-D-thiogalacto pyranoside and incubated at 30 °C and 800 rpm overnight. Bacteria were pelleted by centrifugation for 20 min at $3220 \times g$ and 4 °C.

For the ELISA, 100 ng of antigen was coated on 96-well MTPs (High binding, Greiner) in PBS (pH 7.4) overnight at 4 °C. After coating, the wells were blocked with 2% MPBST for 1 h at RT, followed by three washing steps with $H_2O$ and 0.05% Tween20. Supernatants containing secreted monoclonal scFv were mixed with 2% MPBST (1:2) and incubated onto the antigen-coated plates for 1 h at 37 °C followed by three $H_2O$ and 0.05% Tween20 washing cycles. Bound scFv were detected using murine mAb 9E10, which recognizes the C-terminal c-myc tag (1:50 diluted in 2% MPBST) and a goat anti-mouse serum conjugated with HRP (A0168, Sigma) (1:42,000 dilution in 2% MPBST). Bound antibodies were visualized with TMB substrate (20 parts TMB solution A (30 mM potassium citrate; 1% (w/v) citric acid (pH 4.1)) and 1 part TMB solution B (10 mM TMB; 10% (v/v) acetone; 90% (v/v) ethanol; 80 mM $H_2O_2$ (30%))) were mixed. After stopping the reaction by addition of 1 N $H_2SO_4$, absorbance at 450 nm with a 620-nm reference was measured in an ELISA plate reader (Epoch, BioTek). Monoclonal binders were sequenced and analyzed using VBASE2 (www.vbase2.org)[74], and possible glycosylation positions in the CDRS were analyzed according to Lu et al.[75].

**Inhibition of S1-S2 binding to ACE2 expressing cells using MacsQuant**. The inhibition tests in cytometer on Expi293F cells were performed based on the protocol for "validation of spike protein binding to ACE2" (see above) but only binding to S1-S2-His and RBD-mFc antigen (High Five cell produced) was analyzed. The assay was done in two set-ups. In the first set-up, 50 nM antigen was incubated with min. 1 µM of different scFv-Fc and the ACE2-expressing cells. The resulting median antigen fluorescence of GFP-positive living single cells was measured. For comparison of the different scFv-Fc, first the median fluorescence background of cells without antigen was subtracted, second it was normalized to the antigen signal where no antibody was applied. All scFv-Fc showing an inhibition in this set-up were further analyzed by titration (max. 1500–4.7 nM) on S1-S2-His (High Five cell produced), respectively, on RBD-mFc (max. 100–0.03 nM). The IC50 was calculated using the equation $f(x) = \text{Amin} + (\text{Amax} - \text{Amin})/(1 + (x0/x)^h)^s$ and parameters from Origin Pro 2019. In addition, pairwise combinations (max. 750 nM of each scFv-Fc) of the different inhibiting scFv-Fc were tested.

**Dose-dependent binding of the antibodies (scFv-Fc or IgG format) in titration ELISA**. ELISA were essentially performed as described above in "Screening of monoclonal recombinant binders using E. coli scFv supernatant." For titration ELISA, the purified scFv-hFc were titrated from 3.18 µg/mL–0.001 ng/mL on 30 ng/ well of the following antigens: S1-S2 (High Five cell produced), RBD-mFc (High Five cell produced), S1-mFc (High Five cell produced), and TUN219-2C1-mFc (as control for unspecific Fc binding). In addition, all scFv-hFc were also tested only at the highest concentration (3.18 µg/mL) for unspecific cross-reactivity on Expi293F cell lysate ($10^4$ cells/well), bovine serum albumin (BSA; 1% w/v), and lysozyme. ScFv-hFc or IgG were detected using goat-anti-hIgG(Fc)-HRP (1:70,000, A0170, Sigma). Titration assays were performed using 384-well MTPs (Greiner) using Precision XS microplate sample processor (BioTek), EL406 washer dispenser (BioTek), and BioStack Microplate stacker (BioTek). EC50 were calculated with GraphPad Prism Version 6.1, fitting to a four-parameter logistic curve. The binding of antibodies to S1 subunit His-tagged constructs containing mutations in the RBD region and the cross-reactivity to spike proteins of other coronaviruses was tested as described above. S1-HIS proteins from SARS-CoV-2 (expressed in HEK cells), SARS-CoV-1, MERS, HCoV HKU1, HCoV NL63, and HCoV 229E were acquired commercially (Sino Biologicals products 40591-V08H, 40150-V08B1, 40069-V08H, 40021-V08H, 40601-V08H, 40600-V08H).

**SARS-CoV-2 neutralization in cell culture**. VeroE6 cells (ATCC CRL-1586) were seeded at a density of $6 \times 10^4$/well onto cell culture 96-well plates (Nunc, Cat. #167008). Two days later, cells reached 100% confluence. For neutralization, antibodies (1 µg/mL final concentration) were mixed with the virus inoculum (250 pfu/well), using strain SARS-CoV-2/Münster/FI110320/1/2020 (kind gift of Stephan Ludwig, University of Münster, Germany), in 100 µL full VeroE6 culture medium (Dulbecco's modified Eagle's medium, 10% fetal calf serum, 2 mM glutamine, penicillin, streptomycin) in technical quadruplicates or sixfold replicates

and incubated for 1 h at 37 °C. Then cells were overlaid with the antibody/virus mix and phase-contrast images were taken automatically using a Sartorius Incu-Cyte S3 (×10 objective, 2-h image intervals, 4 images per well) housed in a HeraCell 150i incubator (37 °C, 100% humidity, 5% $CO_2$). Image data were quantified with the IncuCyte S3 GUI tools measuring the decrease of confluence concomitant with the CPE of the virus in relation to uninfected controls and controls without antibody and analyzed with GraphPad Prism 8. Given is the median of the inhibition.

For titration, antibodies were diluted in $1/\sqrt{10}$ steps and mixed with a fixed inoculum of SARS-CoV-2 (100–150 pfu) in a total volume of 500 µL of Vero E6 medium. After 1 h incubation at 37 °C, cells were infected with the antibody/virus mix, incubated for 1 h, and then overlaid with Vero E6 medium containing 1.5% methyl-cellulose. Three days postinfection, wells were imaged using a Sartorius IncuCyte S3 (×4 objective, whole-well scan) and plaques were counted from these images.

**Cloning and production of IgG**. For IgG production, selected antibodies were converted into the human IgG1 format by subcloning of VH in the vector pCSEH1c (heavy chain) and VL in the vector pCSL3l/pCSL3k (light chain lambda/kappa)[76], adapted for Golden Gate Assembly procedure with Esp3I restriction enzyme (New England Biolabs). Expi293F (Thermo Fisher Scientific) cells were transfected with 12.5 µg of both vectors in parallel in a 1:1 ratio. For production, the transfected Expi293F cells were cultured in chemically defined medium F17 (Thermo Fisher Scientific) supplemented with 0.1% pluronic F68 (PAN-Biotech) and 7.5 mM L-glutamine (Merck) for 7 days. A subsequent protein A purification was performed as described above.

**Affinity determination by SPR**. The antibody-binding properties were analyzed at 25 °C on a Biacore™ 8 K instrument (GE Healthcare) using 10 mM HEPES pH 7.4, 150 mM NaCl, 3 mM EDTA, and 0.005% Tween-20 as running buffer. SARS-CoV2 RBDs, wild type and different mutants, or full S-protein, were immobilized on the surface of a CM5 chip through standard amine coupling. Increasing concentration of antibodies (6.25–12.5–25–50–100 nM) were injected using a single-cycle kinetics setting and analyte responses were corrected for unspecific binding and buffer responses. Curve fitting and data analysis were performed with the Biacore™ Insight Evaluation Software (version 2.0.15.12933).

**Analysis of binding to RBD mutants by protein array**. Two nanoliters of the proteins were printed as quadruplicates onto PDITC-coated bScreen slides with a pitch of 700 µm using a SciFlexArrayer (Scienion AG) in non-contact mode. The following proteins were spotted on the array: S1-S2, SARS-CoV-2, High5 (this work); S1, SARS-CoV-2, HEK293 (Sino Biological 40591-V08H); S1, SARS-CoV-2, Baculovirus (Sino Biological 40591-V08B1); S1-RBD, SARS-CoV-2, HEK293 (Sino Biological 40592-V08H); S1-humFc, SARS-CoV-2, HEK293 (Sino Biological 40591-V02H); S1S2, SARS-CoV-2, Baculovirus (Sino Biological 40589-V08B1); S2, SARS-CoV-2, Baculovirus (Sino Biological 40590-V08B); S1-RBD mFc, SARS-CoV-2, HEK293 (Sino Biological 40592-V05H); S1-RBD32, SARS-CoV-2, High5 (this work); S1-RBD25, SARS-CoV-2, High5 (this work); S1-RBD22, SARS-CoV-2, High5 (this work); S2, SARS-CoV-2, High5 (this work); S1, SARS-CoV-2, High5 (this work); S1 mFc, SARS-CoV-2, High5 (this work); S1-RBD32 mFc, SARS-CoV-2, High5 (this work); S1(E484K), SARS-CoV-2, High5 (this work); S1(F486V), SARS-CoV-2, High5 (this work); S1(N438K), SARS-CoV-2, High5 (this work); S1 (G458R), SARS-CoV-2, High5 (this work); S1S2(D614G), SARS-CoV-2, High5 (this work); S1-RBD(V367F), SARS-CoV-2, HEK293 (Sino Biological 40592-V08H1); S1-RBD(V483A), SARS-CoV-2, HEK293 (Sino Biological 40592-V08H5); S1-RBD(R408I), SARS-CoV-2, HEK293 (Acro Biosystems SPD-S52H8); S1-RBD (G476S), SARS-CoV-2, HEK293 (Acro Biosystems SPD-C52H4); S1-RBD(N354D, D364Y), SARS-CoV-2, HEK293 (Acro Biosystems SPD-S52H3); S1-RBD(N354D), SARS-CoV-2, HEK293 (Acro Biosystems SPD-S52H5); S1-RBD(W436R), SARS-CoV-2, HEK293 (Acro Biosystems SPD-S52H7); S1-RBD, SARS-CoV-2, Baculovirus (Sino Biological 40592-V08B); S1(D614G), SARS-CoV-2, HEK293 (Sino Biological 40591-V08H3); S1-S2, SARS, HEK293 (Acro Biosystems SPN-S52H5); S1, SARS, HEK293 (Acro Biosystems SIN-S52H5); S1-RBD, SARS, HEK293 (Acro Biosystems SPD-S52H6); S1-RBD, MERS, Baculovirus (Sino Biological 40071-V08B1); S1, MERS, HEK293 (Sino Biological 40069-V08H); N, MERS, Baculovirus (Sino Biological 40068-V08B); S1, HCoV-229E, HEK293 (Acro Biosystems SIN-V52H4); S1, HCoV-NL63, HEK293 (Acro Biosystems SIN-V52H3); S1, HCoV-HKU1, HEK293 (Sino Biological 40602-V08H); Streptavidin Cy5 (Thermo Fisher 434316); bBSA, (Thermo Fisher 29130), BSA (Carl Roth GmbH 8076.4). Protein concentration was adjusted to 200 µg/mL, with the following exceptions: S1-RBD (E484K) 50 µg/mL, S1-RBD(F486V) 100 µg/mL, S1-RBD(N439K) 138 µg/mL, S1-RBD(G485R) 91 µg/mL. A bscreen was used for label-free measurement of antibody binding to the protein arrays. PBS BSA (1 mg/mL) was used as sample as well as washing buffer. The flow rate was set to 3 µL/s. The measurement was performed in 3 steps—first step: blocking solution (50% PBS BSA 1 mg/mL, 50% Superblock (Thermo Scientific—Cat. No.: 37515)); second step: neutralizing SARS-CoV-2 antibody 8 µg/mL; third step: goat anti-human Alexa 546 antibody 5 µg/mL (Invitrogen—Cat. No.: A-21089). Each step consisted of 150 s baselining, 333 s association, 300 s dissociation. The label-free signals of the association phase of the

anti-human step were used for data generation. This was done by subtracting the signal mean value from 30 to 20 s before the association phase from the signal mean value from 20 to 30 s after the association phase.

**Antibody structures and computational docking studies**. The antibody structures were modeled according to the canonical structure method using the RosettaAntibody program[77] as previously described[78] and docked to the experimental structure of SARS-CoV-2 S protein RBD (PDBid: 6M17)[6].

Docking was performed using the RosettaDock 3.12 software[79] as previously described[80]. Briefly, each antibody was manually placed with the CDR loops facing the RBD region containing the residues identified by the peptide mapping experiment. The two partners were moved away from each other by 25 Å and then brought together by the computational docking algorithm, obtaining thousands of computationally generated complexes (typically 15,000). The antibody/RBD complexes were structurally clustered and then selected according to the scoring function (an estimate of energetically favorable solutions) and agreement with the peptide mapping data. Selected complexes were further optimized by a docking refinement step and molecular dynamics simulations.

The MD simulations were performed using GROMACS[81] with standard MD protocol: antibody/antigen complexes were centered in a triclinic box, 0.2 nm from the edge, filled with SPCE water model and 0.15 M Na+Cl− ions using the AMBER99SB-ILDN protein force field; energy minimization followed. Temperature (298 K) and pressure (1 Bar) equilibration steps of 100 ps each were performed. Five hundred-nanosecond MD simulations was run with the above-mentioned force field for each protein complexes. MD trajectory files were analyzed after removal of periodic boundary conditions. The overall stability of each simulated complex was verified by root mean square deviation, radius of gyration, and visual analysis according to standard procedures.

**Reporting summary**. Further information on research design is available in the Nature Research Reporting Summary linked to this article.

## Data availability

The authors declare that the data supporting the findings of this study are available within the paper and its supplementary information files. All requests for resources and reagents should be directed to the Lead Contact author (m.hust@tu-bs.de). This includes antibodies, plasmids, and proteins. All reagents will be made available on request after completion of a Material Transfer Agreement. Source data are provided with this paper.

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

## Acknowledgements
We kindly acknowledge the support of the European Union for the ATAC ("antibody therapy against corona," Horizon2020 number 101003650) consortium and the MWK Niedersachsen (14-76103-184 CORONA-2/20). L.V. gratefully acknowledges support from SNF grant 31003A_182270 and Lions Club Monteceneri. Work was also supported by the Deutsche Forschungsgemeinschaft (DFG), grant KFO342 to L.B. and S.L. We would like to highlight the passion and motivation of the complete team working on this topic in this special time. We are deeply grateful to Adelheid Langner, Andrea Walzog, Bettina Sandner, Cornelia Oltmann, and Wolfgang Grassl for constant help and support.

## Author contributions
F.B., G. Roth, S.D., L.V., L.Č.-Š., M.S., and M.H. conceptualized the study. F.B., D.M., N.L., S.S., U.R., L.S., P.A.H., R.B., M.R., K.-T.S., K.D.R.R., P.R., K.E., Y.K., D.S., M.P., S.Z.-E., J.W., N.K., T.H., M.B., M.G., S.D.K., G. Roth, and M.S. performed and designed experiments. F.B., D.M., N.L., S.S., U.R., L.S., P.K., G. Roth, L.V., L.Č.-Š., M.S., and M.H. analyzed data. S.L. and L.B. provided material. S.D., L.V., L.Č.-Š., and M.H. conceived the funding. P.K., E.V.W., G. Russo, A.K., V.F., S.D., and M.S. advised on experimental design and data analysis. F.B., S.D., G. Roth, L.V., L.Č.-Š., M.S., and M.H. wrote the manuscript.

## Funding

## Competing interests
The authors declare a conflict of interest. The authors F.B., D.M., N.L., S.S., P.A.H., R.B., M.R., K.T.S., K.D.R.R., S.Z.-E., M.B., V.F., S.T., M.S. and M.H. submitted a patent application on blocking antibodies against SARS-CoV-2. The patent was licensed to CORAT Therapeutics GmbH, a company founded by YUMAB GmbH. S.D. and M.H. are shareholders of YUMAB GmbH.
