## [Peer Review File · Nature Communications]

Reviewer comments first round:

Reviewer #1 (Remarks to the Author):

The authors provide data on the selection of anti-spike glycoprotein SARS-CoV-2 scFv and their use as scFv-Fc fusion proteins to identify molecules with neutralizing activity. This study complements work on generating neutralizing antibodies from B-cells of convalescent patients, but also using other approaches such as phage display of antibody or e.g. nanobody libraries. It, once again, demonstrates that antibodies with potentially therapeutic value can be obtained from antibody phage libraries without the need for immunization or the use of patient-derived materials. Interestingly, a diverse set of antibodies binding to different regions of the RBD could be identified, further demonstrating the versatility of the approach. The work is certainly of great interest to the field.

further comments:

- 1) please use correct name for the virus, SARS-CoV-2, and not SARS2, as used in the abstract.
- 2) the last sentence of the first paragraph in the introduction seems to be incomplete.
- 3) please add in the discussion that most (if not all) antibodies from Covid-19 patient were obtained from convalescent patients.
- 4) please provide a rationale why different expression systems were used and explain why expression in mammalian cells was not optimal, although the virus is naturally produced in human cells.
- 5) please explain if the furin site was also present in S1-hFc produced in Expi293F.
- 6) please provide information on the quality of the scFv-Fc fusion proteins used in the study. Did they retain antigen binding and specificity (not described in the third results paragraph).
- 7) very importantly, authors should include data on possible cross-reactivity with spike protein from other human coronaviruses, e.g. SARS, MERS, and others such as OC43, 229E, HKU-1, and NL63.
- 8) Because recombinant antigen is available, authors should provide affinity data of their most promising candidates.
- 9) the authors describe synergistic effects of combinations of their scFv-Fc fusion proteins. Based on their epitope mapping data, they should discuss if this results from binding to non-overlapping epitopes.
- 10) heading of 7th results paragraph should read "Most, but not all, of the ...
- 11) Please provide information or discuss why the most promising candidates were not converted into full IgGs for further testing, which would be much closer to the real situation and might also affect activities. The authors themselves published previously that conversion of an scFv to a Fab might affect antigen binding.
- 12) The discussion is rather descriptive and repeats many of the findings without an in-depth discussion. They should focus on comparing their antibodies to published antibodies, including combinations of antibodies such as the Regeneron Fabs, and should further focus on translational aspects and possible hurdles.
- 13) it would be more appropriate to use scFv-Fc instead of antibodies to avoid the impression that IgGs were tested.

14) the data shown lack a statistical analysis. How many repetitions of experiments were performed and were observed differences statistically significant. Especially, authors talk about significant effects in the discussion (page 14), without providing a statistical analysis.

Reviewer #2 (Remarks to the Author):

Review of Bertoglio et al. (Nat Comm.)

Title: SARS-CoV-2 neutralizing human recombinant antibodies selected from pre-pandemic healthy donors binding at RBD-ACE2 interface

Summary of results

The authors demonstrated the ability to generate human recombinant antibodies against the Spike antigen of SARS-CoV-2 by screening universal phage libraries expressing naive human antibody genes with CoV-2-specific S1 protein. This study initially selected a total of 309 unique human antibodies against S1. Seventeen of these antibodies were then selected that inhibited the binding of the spike protein to the ACE2 receptor and to ACE2-expressing cells, and blocked SARS-CoV-2 infection of VeroE6 cells. All mabs in this subset were directed against the RBD, and peptide binding assays mapped these to 3 separate sites in the RBD. These antibodies bound to the RBD and blocked spike binding to cells expressing ACE2 with different EC50s, and all of the antibodies neutralized infection of VeroE6 cells by live SARS-Cov-2 virus.

Novelty of results

The novelty of this result is the efficiency of the naive library to produce useful antibodies, which could be important in cases of novel infectious agents where immune patients are not readily available, and therefore more typical sources of antibodies are not available.

Critique of results

There have been many reports of the isolation of human monoclonal antibodies against CoV-2 S1 and RBD from memory B cells of convalescent CoV-2 patients. In most cases, the sequences of the isolated antibodies closely resembled germline sequences, so it is not surprising that similar antibodies were derived from naive antibody libraries. Nonetheless, this report is of interest because of the apparent ease in which functional antibodies were isolated from a CpV-2 naive antibody library.

However, to demonstrate the utility of this system it is necessary to fully document the specificity and functional potency of the resulting antibodies, and to see how these compare to the typical antibodies that have been isolated from more standard sources; i.e., infected humans and immunized humans and animals.

1- The authors show at least some characterization of antibody sequence information of the selected antibodies, to better understand the diversity and complexity of the selected mabs. At the minimum, they should show the related germ-line sequences, % somatic mutation and H and L chain CDR3 sequences for each mab.

2- Neutralization results are shown only for one concentration of antibody 1 µg/ml (~100 nM), and are expressed as the % neutralization seen at that concentration. This does not provide a useful indication of the neutralization potency or allow comparison to activities of standard mAbs described elsewhere. At the very least, accurate IC50s should be determined for all of the neutralizing antibodies.

3- As indicated in Table 2, only 17/110 unique antibodies expressed in scFv-Fc possessed inhibiting activities in an ACE2 binding assay, and those antibodies were selected for follow-up studies. It is well known that only a fraction of known neutralizing antibodies block ACE2 binding, and it is important to learn more about the functional activities of antibodies that bind to site in the S1 protein outside of the RBD.

How did the binding affinities of the selected antibodies compare to those that were not selected, and what is known about the epitope specificity of the non-selected population? Did these also

bind to RBD, or were they localized elsewhere in S1? Did any of these possess neutralizing activities that were not necessarily related to inhibition of ACE2 binding, and what were their neutralization potencies?

Other comments

The description of the number of selections performed and number of antibodies isolated described on pages 7 and 8 are not consistent with the totals listed in Table 2 and elsewhere in the paper. This should be corrected.

Reviewer #3 (Remarks to the Author):

The manuscript by Bertoglio et al focuses on a very hot topic given the lack of drugs tailored on SARS-CoV-2. Identification and characterization of monoclonal antibodies endowed with neutralizing activity against SARS-CoV-2 certainly represent an important biomedical research branch. However, despite the importance of the topic treated by the authors, the manuscript presents several important limitations.

Major drawbacks:

Why have the authors produced all spike formats of SARS-CoV-2 in two eukaryotic systems? As stated by authors in the Discussion, most antibodies were obtained by performing panning on antigens produced on insect cells compared to few antibodies obtained using protein expressed by mammalian cells. Despite similar glycosylation pattern of the two systems, possible biases should be carefully evaluated. Moreover, different glycosylation patterns have been recently described for different spike formats (i.e. S0 vs S1 or S2) produced on the same system. Also, why did they discarded antibodies selected against S1-hFc produced in Hexpi 293 F? Authors must also explain why they used two S1 forms (wild type vs furin abrogated format).

Under the light of the extreme importance of identifying anti-SARS-CoV-2 antibodies in humans, the features of both donors and libraries should be clearly and exhaustively described in the manuscript. Also, the similarity to the germlines should be added and reported in the main text.

One of the main concerns on the manuscript is the lack of an accurate analysis of the IC50 of the different antibodies against SARS-CoV-2. At this purpose author tested their antibodies at a single concentration. This is not enough for calculating the IC50 value for the single molecules, therefore, they should test a curve of concentration spanning from the concentration fully inhibiting the infection to ineffective concentration. Moreover, the authors should perform the virus titration by using also different methods for accurately determine the MOI effectively used in the experiments. Last, at least two MOI of virus should be used for precisely assessing the neutralizing activity of the antibodies. Viral RNA amount should be also evaluated for each condition.

Regarding the testing of synergisms: the whole description of experiments results confusing and the experiments are not described linearly (even the name of the receptor is not present in the paragraph describing the results). Moreover, the authors should analyse synergy data by using different mathematical approaches.

It is not clear at all how authors calculated EC50 from the ELISA binding assay

Regarding the epitope mapping:

The use of 15 mer library is not enough and exhaustive at all for the epitope mapping given the conformational nature of epitopes recognized by most human antibodies. Further experiments, such as alanine scanning are mandatory for possibly corroborating what observed by peptide panning techniques.

The computational pipeline is poorly described and possibly biased by the constraints introduced by the initial set-up of the system.

The authors mention MD simulations without providing any details and therefore is not clear at all

why they performed it.

The authors, should mention, at least in the Supplementary, the free energy of binding of the selected complexes and provide a comparison with the binding observed in ELISA (this step would be a control of in silico analysis)

Reviewer 1:

„1) please use correct name for the virus, SARS-CoV-2, and not SARS2, as used in the abstract.“

We corrected this typo.

„2) the last sentence of the first paragraph in the introduction seems to be incomplete.“

We corrected this sentence.

„3) please add in the dicussion that most (if not all) antibodies from Covid-19 patient were obtained from convalescent patients.“

We added this point.

„4) please provide a rational why different expression systems were used and explain why expression in mammalian cells was not optimal, although the virus is naturally produced in human cells.“

Beginning of February 2020, it was not clear which production hosts were best suited for SARS-CoV-2 antigen expression. We used both production systems in parallel to increase the chance of getting functional material timely, in reasonable amounts during the pandemic situation. The RBD expression is working well in both systems, but S1 and S1-S2 are produced with much higher yields in insect cells. The functionality was tested on recombinant ACE2 and on cells expressing ACE2.

„5) please explain if the furin site was also present in S1-hFc produced in Expi293F.“

Walls et al. 2020 Cell (published in March) reported: „We identified the presence of an unexpected furin cleavage site at the S1/S2 boundary of SARS-CoV-2 S, which is cleaved during biosynthesis—a novel feature setting this virus apart from SARS-CoV and SARSr-CoVs.“

We also first tried to produce the S1-hFc with the original amino acid sequence including the furin site in mammalian and insect cells and were able to purify the cleaved material by SEC, but this resulted in very low amounts. For this reason, we removed the furin site according to Walls et al 2020. We also added an additional supplementary figure (Supplementary data 1) to illustrate all the constructs.

„6) please provide information on the quality of the scFv-Fc fusion proteins used in the study. Did they retain antigen binding and specificity (not described in the third results paragraph).“

Antigen binding and specificity of the scFv-Fc antibodies is described in Figure 3 and all following assays (e.g. inhibition and neutralization assays).

„7) very importantly, authors should include data on possible cross-reactivity with spike protein from other human coronaviruses, e.g. SARS, MERS, and others such as OC43, 229E, HKU-1, and NL63.“
We added binding data to other coronaviruses for the most promising antibody STE73-2E9 (Figure 7). In addition, we also added data for the binding to known RBD mutants for the three inhibiting IgGs (Figure 6).

„8) Because recombinant antigen is available, authors should provide affinity data of their most promising candidates.“

We added affinity measurements for the most promising antibody STE73-2E9 (new Figure 7).

„9) the authors describe synergistic effects of combinations of their scFv-Fc fusion proteins. Based on their epitope mapping data, they should discuss if this results from binding to non-overlapping epitopes.“

Because we now focus on the inhibiting IgGs and their corresponding epitopes, we moved the scFv-Fc combinations assays to the supplementary part and do not deeply discuss these results.

„10) heading of 7th results paragraph should read "Most, but not all, of the ...“

Because of the refined study, this paragraph was completely rewritten in the revision.

„11) Please provide information or discuss why the most promising candidates were not converted into full IgGs for further testing, which would be much closer to the real situation and might also affect activities. The authors itself published previously that conversion of an scFv to a Fab might affect antigen binding.“

We agree with the reviewer and converted the most promising antibodies to the IgG format. These antibodies were now validated by ELISA, inhibition assays and the best IgG was further analyzed (binding to RBD mutants, other coronaviruses, affinity and aggregation behaviour).

„12) The discussion is rather descriptive and repeats many of the finding without an in-depth discussion. They should focus on comparing their antibodies to published antibodies, including combinations of antibodies such as the Regeneron Fabs, and should further focus on translational aspects and possible hurdles.“

With the focus on the IgG STE73-2E9 we improved the discussion.

„13) it would be more appropriate to use scFv-Fc instead of antibodies to avoid the impression that IgGs were tested.“

We now added IgG data and use the nomenclature "scFv-Fc" or "IgG" to unmistakably identify the format whenever necessary.

"14) the data shown lack a statistical analysis. How many repetitions of experiments were performed and were observed differences statistically significant. Especially, authors talk about significant effects in the discussion (page 14), without providing a statistical analysis."

We added this information to the figure legends

Reviewer 2:

"1- The authors show at least some characterization of antibody sequence information of the selected antibodies, to better understand the diversity and complexity of the selected mAbs. At the minimum, they should show the related germ-line sequences, % somatic mutation and H and L chain CDR3 sequences for each mAb."

We added the germinality index in the revised Table 3.

"2- Neutralization results are shown only for one concentration of antibody 1 µg/ml (~100 nM), and are expressed as the % neutralization seen at that concentration. This does not provide a useful indication of the neutralization potency or allow comparison to activities of standard mAbs described elsewhere. At the very least, accurate IC50s should be determined for all of the neutralizing antibodies."

The neutralization assays were performed again with the IgG in a plaque titration assay. STE73-2E9 was further characterized and the IC50 in the neutralization assays was determined.

"3- As indicated in Table 2, only 17/110 unique antibodies expressed in scFv-Fc possessed inhibiting activities in an ACE2 binding assay, and those antibodies were selected for follow-up studies. It is well known that only a fraction of known neutralizing antibodies block ACE2 binding, and it is important to learn more about the functional activities of antibodies that bind to site in the S1 protein outside of the RBD.

How did the binding affinities of the selected antibodies compare to those that were not selected, and what is known about the epitope specificity of the non-selected population? Did these also bind to RBD, or were they localized elsewhere in S1? Did any of these possess neutralizing activities that were not necessarily related to inhibition of ACE2 binding, and what were their neutralization potencies?"

As a general comment, we have seen that the vast majority of the selected and produced scFv-Fcs were RBD binders, with extremely rare exceptions. We agree that the analysis of the scFv-Fc which does not inhibit directly Spike binding to ACE2 would be certainly a rewarding research project on its own, but it is far beyond the focus of this work.

"The description of the number of selections performed and number of antibodies isolated described on pages 7 and 8 are not consistent with the totals listed in Table 2 and elsewhere in the paper. This should be corrected."

Thanks for pointing that out, we corrected this point.

Reviewer 3:

„Why have the authors produced all spike formats of SARS-CoV-2 in two eukaryotic systems? As stated by authors in the Discussion, most antibodies were obtained by performing panning on antigens produced on insect cells compared to few antibodies obtained using protein expressed by mammalian cells. Despite similar glycosylation pattern of the two systems, possible biases should be carefully evaluated. Moreover, different glycosylation patterns have been recently described for different spike formats (i.e. S0 vs S1 or S2) produced on the same system. Also, why did they discarded antibodies selected against S1-hFc produced in Hexpi 293 F? Authors must also explain why they used two S1 forms (wild type vs furin abrogated format).“

In February 2020, it was not clear which production hosts were best suited for SARS-CoV-2 antigen expression. We used both production systems in parallel to increase the chance of getting functional material timely, in reasonable amounts during the pandemic situation. RBD expression is working well in both systems, but S1 and S1-S2 are produced with much higher yields in insect cells. The functionality was tested on recombinant ACE2 and on cells expressing ACE2. We also first tried to produce the S1-hFc with the original amino acid sequence including the furin site in mammalian and insect cells and were able to purify the cleaved material by SEC, but this resulted in very low amounts. For this reason, we removed the furin site according to Walls et al 2020. We also added an additional supplementary figure (Supplementary Data 1) to illustrate all the constructs.

„Under the light of the extreme importance of identifying anti-SARS-CoV-2 antibodies in humans, the features of both donors and libraries should be clearly and exhaustively described in the manuscript. Also, the similarity to the germlines should be added and reported in the main text.“

We added the germinality index information in Table 3.

„One of the main concerns on the manuscript is the lack of an accurate analysis of the IC50 of the different antibodies against SARS-CoV-2. At this purpose author tested their antibodies at a single concentration. This is not enough for calculating the IC50 value for the single molecules, therefore, they should test a curve of concentration spanning from the concentration fully inhibiting the infection to ineffective concentration. Moreover, the authors should perform the virus titration by using also different methods for accurately determine the MOI effectively used in the experiments. Last, at least two MOI of virus should be used for precisely assessing the neutralizing activity of the antibodies. Viral RNA amount should be also evaluated for each condition.“

We now produced the most promising antibodies as IgG and tested these antibodies by titration in a SARS-CoV-2 plaque assay to determine the IC50.

„Regarding the testing of synergisms: the whole description of experiments results confusing and the experiments are not described linearly (even the name of the receptor is not present in the paragraph describing the results). Moreover, the authors should analyse synergy data by using different mathematical approaches.“

We specified that this assay was performed on S1-S2. However, we moved the antibody combinations to the Supplementary data, because we focused additional experiments on IgGs and RBD mutants.

„It is not clear at all how authors calculated EC50 from the ELISA binding assay“

EC50 were calculated with GraphPad Prism Version 6.1, fitting to a four-parameter logistic curve. We described the EC50 calculation in the Material and Methods part. We also added this information to the figure legends.

„Regarding the epitope mapping: The use of 15 mer library is not enough and exhaustive at all for the epitope mapping given the conformational nature of epitopes recognized by most human antibodies. Further experiments, such as alanine scanning are mandatory for possibly corroborating what observed by peptide panning techniques.“

We agree with the reviewer that a 15mer library is not a perfect solution to determine the epitopes. For this reason, we now analyzed the binding to RBD mutants with three different approaches (ELISA, SPR and protein arrays) and used this data set to corroborate docking modelling studies to predict the epitopes of the three antibodies which show inhibition as IgG.

„The computational pipeline is poorly described and possibly biased by the constraints introduced by the initial set-up of the system.“

We improved the description of the computational pipeline in the M&M part.

„The authors mention MD simulations without providing any details and therefore is not clear at all why they performed it.“

We added the information about the MD simulations in the M&M part.

Reviewer comments second round:

Reviewer #1 (Remarks to the Author):

The authors have sufficiently addressed most of the questions raised by the reviewers. Especially, it is appreciated that data on IgG molecules have been added to the main section. That authors do not want to show sequences of the antibodies or the CDRs is understandable in light of a patent application.

However, information on the statistics is still missing in most of the results and figure legends, although this is stated in the response letter. I.e. authors should indicate number of replicates (n) and indicate if e.g. mean +/- SD or SEM is shown in the Figures (several graphs show error bars). Furthermore, some graphs do not show any error bars and I am wondering if the results are based on a single measurement, which would scientifically not be very robust.

Reviewer #2 (Remarks to the Author):

1. There was a request by several of the reviewers that "at least some characterization of antibody sequence information of the selected antibodies, to better understand the diversity and complexity of the selected mabs. At the minimum, they should show the related germ-line sequences, % somatic mutation and H and L chain CDR3 sequences for each mab."

The authors replied "We added the germinality index in the revised Table 3." This provides limited information, and does not really respond to the critique, and does not provide insight into the relationships between the various CDR regions. At least the H and L chain CDR3 should be included.

2- There was a request from several reviewers that more quantitative neutralization data be provided, so that the utility of these antibodies could be compared to those of antibodies isolated from convalescent patients. In response the authors provides % neutralization achieved at 1 µg/ml of the scFv-Fvs and the IC50 value for three IgGs in a plaque titration assay. This data is shown in Fig. 7, and reports an IC50 of 0.43 nM (equivalent to ~64 ng/ml) for STE73-2E9, The IC50 values for the two other IgGs tested are not reported, but from the curves it is clear that these were higher or undeterminable.

This response does not address the issue for the other reagents described, for which neutralization data is reported for the scFv-Fv forms only at a single concentration (1 µg/ml), which was saturating for many of the samples (100%). More quantitative data is needed in order to understand the relatively utility of these samples (what is the point of describing them, if they are not being characterized in a useful way?)

The potency of STE73-2E9 is considerably lower than that reported for the best of the natural human mabs to RBD, which can be as low as 1 ng/ml (see Rogers et al. <https://science.sciencemag.org/content/sci/early/2020/06/15/science.abc7520.full.pdf>). This comparison should be discussed, and included in the evaluation of the utility of these reagents and this approach.

This is not consistent with the conclusion on page 15, that "with few exceptions our approach demonstrates that human antibodies with functional properties matching those of the antibodies isolated from convalescent patients can be generated without the necessity to wait for material from COVID-19 infected individuals. Therefore, this strategy offers a very fast additional opportunity to respond to future pandemics". This should be replaced with a more accurate evaluation.

Reviewer 1:

"The authors have sufficiently addressed most of the questions raised by the reviewers. Especially, it is appreciated that data on IgG molecules have been added to the main section. That authors do not want to show sequences of the antibodies or the CDRs is understandable in light of a patent application."

We now provide the sequences of all inhibiting antibodies. We also marked the CDRs in the sequences.

"However, information on the statistics is still missing in most of the results and figure legends, although this is stated in the response letter. I.e. authors should indicate number of replicates (n) and indicate if e.g. mean +/- SD or SEM is shown in the Figures (several graphs show error bars). Furthermore, some graphs do not show any error bars and I am wondering if the results are based on a single measurement, which would scientifically not be very robust."

We added the respective informations to the figure legends.

Reviewer 2:

"1. There was a request by several of the reviewers that "at least some characterization of antibody sequence information of the selected antibodies, to better understand the diversity and complexity of the selected mabs. At the minimum, they should show the related germ-line sequences, % somatic mutation and H and L chain CDR3 sequences for each mab. The authors replied "We added the germinality index in the revised Table 3." This provides limited information, and does not really respond to the critique, and does not provide insight into the relationships between the various CDR regions. At least the H and L chain CDR3 should be included."

We added V-genes sequences of all 17 inhibiting scFv-Fc as supplementary Data 4. We also marked the CDRs in the sequence.

"2- There was a request from several reviewers that more quantitative neutralization data be provided, so that the utility of these antibodies could be compared to those of antibodies isolated from convalescent patients. In response the authors provides % neutralization achieved at 1 µg/ml of the scFv-Fvs and the IC50 value for three IgGs in a plaque titration assay. This data is shown in Fig. 7, and reports an IC50 of 0.43 nM (equivalent to ~64 ng/ml) for STE73-2E9, The IC50 values for the two other IgGs tested are not reported, but from the curves it is clear that these were higher or undeterminable."

We added the IC50 values of the other two tested IgGs. The IC50 of STE73-9G3 was validated with a second plaque assay with ~150 pfu resulting in an IC50 of 0.41 nM (new figure 7B).

"This response does not address the issue for the other reagents described, for which neutralization data is reported for the scFv-Fv forms only at a single concentration (1 µg/ml), which was saturating for many of the samples (100%). More quantitative data is needed in order to understand the relative utility of these samples (what is the point of describing them, if they are not being characterized in a useful way?)"

This live virus neutralization assay was used as a rapid screening tool to decide which inhibiting scFv-Fc will be subcloned as full IgG for further characterization. Therefore, no further effort was put into quantification of the activity of the intermediate scFv-Fc format, as this characterization was done in detail anyway later for the IgG variants, which represent the only format relevant for the clinical development. Detailed quantitative functional data is presented for all of the final IgGs. We added a comment on the utility of these samples to make this more clear.

"The potency of STE73-2E9 is considerably lower than that reported for the best of the natural human mabs to RBD, which can be as low as 1 ng/ml (see Rogers et al. <https://science.sciencemag.org/content/sci/early/2020/06/15/science.abc7520.full.pdf>). This comparison should be discussed, and included in the evaluation of the utility of these reagents and this approach."

Indeed, since we submitted this paper at the beginning of June, several other good anti-RBD antibodies were published. It should be kept in mind that biological activities very much depend on the respective assay setup, and comparison of IC50 just by the number may be misleading. For example, the antibody CC6.33 (Rogers et al.) is neutralizing pseudovirus SARS-CoV-2 with an IC50 of 1 ng/mL which makes the comparison difficult, because we used authentic SARS-CoV-2. We therefore added a detailed discussion on the IC50 values.

"This is not consistent with the conclusion on page 15, that "with few exceptions our approach demonstrates that human antibodies with functional properties matching those of the antibodies isolated from convalescent patients can be generated without the necessity to wait for material from COVID-19 infected individuals. Therefore, this strategy offers a very fast additional opportunity to respond to future pandemics". This should be replaced with a more accurate evaluation."

With SARS-CoV-2 the whole world invested an unprecedented amount of resources towards a single goal. This is not common for other diseases and infective outbreaks. The fact that research groups and companies were able to rapidly recruit many individuals and quickly find potentially good antibodies should not be held as standard for every situation. The antibody discovery from naive libraries described in this paper offers at least one valid approach in a pandemic situation. We modified this part of the discussion with a more accurate statement.

Reviewer comments third round:

Reviewer #1 (Remarks to the Author):

All issues raised by the reviewer have sufficiently been addressed. Authors added information on the number of repetitions and error bars to the figure legends, i.e. Fig. 2 to 7, and the supplementary figures. Although some measurements were only performed as individual measurements, the final neutralization assays, e.g. shown in Fig.7B are based on triplicates. With the information provided the reader can now deduce experimental robustness.

Reviewer #2 (Remarks to the Author):

The authors adequately addressed most of the critiques of the reviewers. However, one issue remains.

In the Discussion, the authors state

"Our approach demonstrates that human antibodies with functional properties MATCHING THOSE OF THE ANTIBODIES ISOLATED FROM CONVALESCENT PATIENTS can be generated without the necessity to wait for material from COVID-19 infected individuals".

As pointed out by this reviewer, this is not an accurate statement, since their best Ab is at least 50-fold weaker than the best of the mAbs isolated from convalescent patients. It is therefore necessary to amend this statement so that it accurately reflects the data.

The authors recognise this point, and rationalise that "With SARS-CoV-2 the whole world invested an unprecedented amount of resources towards a single goal. This is not common for other diseases and infective outbreaks. The fact that research groups and companies were able to rapidly recruit many individuals and quickly find potentially good antibodies should not be held as standard for every situation. Therefore, this strategy offers a very fast additional opportunity to respond to future pandemics."

These are all valid points, but they do not make their inaccurate statement true.

The further address this issue later in the Discussion (see below), and they make the point that in some cases different assays were used (pseudoviruses rather than live virus), and therefore they cannot directly compare these values. Although in some cases significant differences in potencies were reported for the two assays, most studies do not show much differences. Even assuming that their point is valid, this does not make their statement true.

"The best neutralizing antibody STE73-2E9 showed an IC₅₀ of 0.41 nM. Cao et al. 18 reported an IC₅₀ of 33 ng/mL (~0.22 nM) for BD368-2 in a comparable live virus plaque assay. Other publications also reports better IC₅₀ efficacies, e.g. Kreye et al. for CV07-209 with 3.1 ng/mL (~0.02 nM) or Rogers et al. 61 for CC6.33 with 1 ng/mL but the different assays are often not directly comparable, e.g. Rogers et al. used SARS-CoV-2 pseudovirus instead of authentic virus. In addition, these antibodies are derived from COVID-19 patients and not from a naive antibody gene library".

I would suggest that the offending statement be modified to something like the following: "Our approach demonstrates that human antibodies with effective functional properties can be generated without the necessity to wait for material from COVID-19 infected individuals". They can then discuss the fact that their IC₅₀s are not as potent as those reported for the best ones isolated from convalescent patients, and provide their explanation for why this is so. This would not invalidate the utility of their approach, but would more accurately reflect the reality of the situation.

Reviewer 2:

"Our approach demonstrates that human antibodies with functional properties MATCHING THOSE OF THE ANTIBODIES ISOLATED FROM CONVALESCENT PATIENTS can be generated without the necessity to wait for material from COVID-19 infected individuals".

As pointed out by this reviewer, this is not an accurate statement, since their best Ab is at least 50-fold weaker than the best of the mAbs isolated from convalescent patients. It is therefore necessary to amend this statement so that it accurately reflects the data. The authors recognise this point, and rationalise that "With SARS-CoV-2 the whole world invested an unprecedented amount of resources towards a single goal. This is not common for other diseases and infective outbreaks. The fact that research groups and companies were able to rapidly recruit many individuals and quickly find potentially good antibodies should not be held as standard for every situation. Therefore, this strategy offers a very fast additional opportunity to respond to future pandemics."

These are all valid points, but they do not make their inaccurate statement true. The further address this issue later in the Discussion (see below), and they make the point that in some cases different assays were used (pseudoviruses rather than live virus), and therefore they cannot directly compare these values. Although in some cases significant differences in potencies were reported for the two assays, most studies do not show much differences. Even assuming that their point is valid, this does not make their statement true. "The best neutralizing antibody STE73-2E9 showed an IC₅₀ of 0.41 nM. Cao et al. 18 reported an IC₅₀ of 33 ng/mL (~0.22 nM) for BD368-2 in a comparable live virus plaque assay. Other publications also reports better IC₅₀ efficacies, e.g. Kreye et al. for CV07-209 with 3.1 ng/mL (~0.02 nM) or Rogers et al. 61 for CC6.33 with 1 ng/mL but the different assays are often not directly comparable, e.g. Rogers et al. used SARS-CoV-2 pseudovirus instead of authentic virus. In addition, these antibodies are derived from COVID-19 patients and not from a naive antibody gene library". I would suggest that the offending statement be modified to something like the following: "Our approach demonstrates that human antibodies with effective functional properties can be generated without the necessity to wait for material from COVID-19 infected individuals". They can then discuss the fact that their IC₅₀s are not as potent as those reported for the best ones isolated from convalescent patients, and provide their explanation for why this is so. This would not invalidate the utility of their approach, but would more accurately reflect the reality of the situation."

We agree with the reviewer and changed the statement as suggested. We also added the information, that the antibody STE73-2E9 from a naive antibody gene library has a higher IC50 compared to the antibodies derived from COVID-19 patients.